# A comprehensive model of *Drosophila* epithelium reveals the role of embryo geometry and cell topology in mechanical responses

Mohamad Ibrahim Cheikh[1], Joel Tchoufag[2,3], Miriam Osterfield[1], Kevin Dean[4], Swayamdipta Bhaduri[1], Chuzhong Zhang[5], Kranthi Kiran Mandadapu[2,3], Konstantin Doubrovinski[1]*

[1]Department of Biophysics, University of Texas Southwestern Medical Center, Dallas, United States; [2]Department of Chemical and Biomolecular Engineering, University of California, Berkeley, Berkeley, United States; [3]Chemical Sciences Division, Lawrence Berkeley National Laboratory, Berkeley, United States; [4]Department of Bioinformatics, University of Texas Southwestern Medical Center, Dallas, United States; [5]Department of Material Science and Engineering, University of Texas at Arlington, Arlington, United States

**\*For correspondence:**
Konstantin.Doubrovinski@
UTSouthwestern.edu

**Competing interest:** The authors declare that no competing interests exist.

**Abstract** In order to understand morphogenesis, it is necessary to know the material properties or forces shaping the living tissue. In spite of this need, very few in vivo measurements are currently available. Here, using the early *Drosophila* embryo as a model, we describe a novel cantilever-based technique which allows for the simultaneous quantification of applied force and tissue displacement in a living embryo. By analyzing data from a series of experiments in which embryonic epithelium is subjected to developmentally relevant perturbations, we conclude that the response to applied force is adiabatic and is dominated by elastic forces and geometric constraints, or system size effects. Crucially, computational modeling of the experimental data indicated that the apical surface of the epithelium must be softer than the basal surface, a result which we confirmed experimentally. Further, we used the combination of experimental data and comprehensive computational model to estimate the elastic modulus of the apical surface and set a lower bound on the elastic modulus of the basal surface. More generally, our investigations revealed important general features that we believe should be more widely addressed when quantitatively modeling tissue mechanics in any system. Specifically, different compartments of the same cell can have very different mechanical properties; when they do, they can contribute differently to different mechanical stimuli and cannot be merely averaged together. Additionally, tissue geometry can play a substantial role in mechanical response, and cannot be neglected.

## Editor's evaluation

Using a novel micropipette-based, minimally invasive approach in combination with theoretical and computational analysis, this important work probes tissue mechanics in the *Drosophila* embryo. The authors provide compelling evidence for the applicability of their method, which reveals important differences between the mechanical properties on the apical and basal tissue sides. This work should be of broad interest to scientists studying tissue mechanics, membranes, and developmental processes.

## Introduction

During development, many organs are shaped through epithelial morphogenesis, a process in which epithelial cell sheets actively deform to give rise to complex structures. The early *Drosophila* embryo has been an important system for studying epithelial morphogenesis (*Blankenship et al., 2006*; *Kiehart et al., 2017*; *Martin et al., 2009*), as it is readily amenable to genetic techniques, high-resolution imaging, and physical perturbations. Ventral furrow formation, an early morphogenetic event in which the mesoderm moves from the exterior to the interior of the fly embryo, has been particularly widely studied process which is understood in great genetic and molecular detail (*Ko and Martin, 2020*; *Stathopoulos and Newcomb, 2020*). The physical mechanisms by which the ventral furrow forms have also been widely examined, but these studies have failed to reach a consensus regarding what spatiotemporal patterns of mechanical forces and material properties drive this physical transformation (*Fierling et al., 2022*; *Heer et al., 2017*; *Perez-Mockus et al., 2017*; *Polyakov et al., 2014*; *Rauzi et al., 2013*). A truly predictive mechanical model of ventral furrow formation would require an accurate representation of the patterns of forces and the patterns of material properties in the embryo, since these two factors combined determine the dynamics of tissue movement.

To constrain models of tissue morphogenesis enough to accurately describe the underlying physics, it should suffice to know any two of these: the dynamics of tissue movement, the pattern of forces, and the pattern of material properties. The dynamics of tissue movement can often be easily ascertained from microscopy. Techniques for the measurement of forces in vivo are still quite limited (*Cost et al., 2015*; *Eder et al., 2017*; *Polacheck and Chen, 2016*; *Roca-Cusachs et al., 2017*), so an increasingly productive approach has been to develop techniques for probing material properties of cells (*Wu et al., 2018*). However, these methods have several shortcomings. Optical and magnetic tweezers (*Bambardekar et al., 2015*; *Tanase et al., 2007*) typically generate forces in the piconewton range, such that the resulting displacements do not exceed a few microns. Additionally, those forces are usually applied on a time-scale of seconds. Hence, the force- and time-scales in these measurements tend to be well below the scales characteristic of morphogenesis, which typically unfolds on a time-scale of tens of minutes or hours and results in deformations on the order of tens to hundreds of microns. Since material properties are in general expected to vary depending on the time- and force-scales involved, such measurements may not be relevant for understanding the physical mechanisms underlying morphogenesis. Bendable cantilevers and biological atomic force microscopes (AFM) allow for applying much larger forces over larger times (*Fernández and Ott, 2008*; *Fischer-Friedrich et al., 2014*). However, these techniques are hard to use in living embryos since introducing a probe into an embryo without significantly disrupting the tissue proves challenging. Recently, we and one other lab introduced a measurement technique based on the use of ferrofluid droplets (*Doubrovinski et al., 2017*; *Serwane et al., 2017*). This method allows probing tissue with forces on the order of tens of nanonewtons, but requires a probe that is fairly large (10–20 μm; several-fold larger than a typical cell), thus possibly locally damaging the tissue that is being probed.

Previously, we and others used ferrofluid droplets to measure the viscosity of cytoplasm in the *Drosophila* embryo, which we found to be approximately 1000 cP (*Doubrovinski et al., 2017*; *Selvaggi et al., 2018*). We also demonstrated that elasticity is largely due to F-actin, indicating that the actin-rich cortex is the primary source of elasticity, and we quantified the time-scale of elastic stress relaxation (*Doubrovinski et al., 2017*). However, we were not able to measure the Young's (or elastic) modulus of the cortex. To accomplish this, and to address the shortcomings of the currently available methods, we introduce here a novel measurement technique based on soft bendable polymer-based cantilevers that may be introduced into an embryo to probe tissue mechanics with high spatial precision. These measurements, combined with computational modeling, reveal that response to applied force is dominated by elastic forces and geometric constraints, and that viscous friction between the tissue and its environment is largely irrelevant. Importantly, these combined approaches also allowed us to estimate the elastic modulus of the apical surface of the epithelial cells, and to set a lower bound on the elastic modulus of the apical surface.

## Results

### Flexible cantilevers as a probe for tissue properties

Progress toward establishing the mechanical properties of animal tissue is limited by both the availability of measurements and the development of quantitative models that can realistically account for both the material properties and the geometry of cells. To tackle the first challenge, we developed a novel technique which is inspired by previous measurements on tissue culture cells where a bendable cantilever of glass was used as a force probe (*Desprat et al., 2005*). In these experiments, the deflection of the cantilever is used to quantify the force applied by the tip; in combination with the resulting tissue deformation, a force-displacement relationship for the tissue can be established.

To probe the mechanics of individual embryonic cell edges, we needed to create a novel tool that (1) can be inserted into a single embryonic cell, (2) is sufficiently compliant to appreciably bend under the relevant nanonewton-range forces, and (3) is capable of being visualized within the rather opaque embryonic interior. To meet these needs, we developed a technique for manufacturing a probe consisting of a long stiff glass pipette with a soft, fluorescent polymer tip that acts as a short flexible cantilever (*Figure 1a*). Briefly, a standard injection glass pipette is used as a mold for making a PDMS (polydimethylsiloxane) cantilever. The tip of a glass pipette is filled with an uncured mixture of polymer, crosslinker, and a green variant of BODIPY (boron-dipyrromethene) dye, which enters by capillary action through an opening at the tip. After the polymer is cured, the tip of the glass scaffold is etched away in a solution of hydrofluoric acid to expose the soft PDMS core. This technique is described in further detail in the Materials and methods section and in *Figure 1—figure supplement 1*.

With this probe in hand, we began to study the mechanical properties of cellularization stage *Drosophila* embryos. From fertilization through the earliest several cycles of mitosis, the *Drosophila* embryo is a syncytium, that is a very large cell with multiple nuclei sharing common cytoplasm. Subsequently, in the process of cellularization, membranes extend from the surface of the embryo and protrude between the nuclei, such that the cytoplasm is for the first time partitioned into individual cells (*Campos-Ortega and Hartenstein, 1985*). During cellularization, every newly forming cell is in effect open on its basal side, such that the interior of that cell is connected to the yolk compartment residing in the center of the embryo (see *Figure 1b* for a schematic illustration). *Drosophila* embryos at this stage have become a key model for studying epithelial mechanical properties (*Bambardekar et al., 2015*; *D'Angelo et al., 2019*; *Doubrovinski et al., 2017*), as it forms a basis for understanding the molecular and physical changes that must occur to drive the large-scale tissue morphogenesis that begins shortly afterward (*Blankenship et al., 2006*; *Kiehart et al., 2017*; *Martin et al., 2009*).

To introduce our probe into a cellularization-stage embryo, we use a pulled glass pipette to cut a slit in the embryo on the opposite surface from which we plan to take measurements (*Figure 1b*). Importantly, we found that when a fly embryo is cut along one side, the side that is not cut continues to develop completely normally showing no noticeable defects. We insert the probe through this cut, then through the basal hole of a cell on the far side of the embryo, so that the flexible tip of the cantilever is positioned within a single cell. For consistency, we always make the initial incision on the ventral side of the embryo and probe material properties of cells of the dorsal side. Once the probe is inserted, we use a piezo actuator mounted on the microscope stage to control cantilever movement. As the cantilever is translated parallel to the cellular layer, it pushes against the adjacent cell edge, thus exerting a pulling force on that cell (*Videos 1 and 2*, *Figure 1b*).

Each pulling experiment consists of three phases. First, during the loading phase, the base of the cantilever is translated at a constant velocity of 0.5 μm/s for approximately 60 s. Next, during the holding phase, the cantilever is held in place for approximately 60 s. Finally, in the unloading (or relaxation) phase, the cantilever is rapidly retracted from the interior of the cell, and the cellular layer is then allowed to freely relax. Through this experiment, the bending of the cantilever and the deformation of the cellular layer are imaged for subsequent quantification. For time frames from a typical pulling experiment, see *Figure 1c*.

### Cantilevers allow simultaneous quantification of applied force and tissue deformation

To quantitatively examine displacement information from this type of experiment, we plotted the data in two distinct ways. First, we represented the imaging data from each experiment as a kymograph,

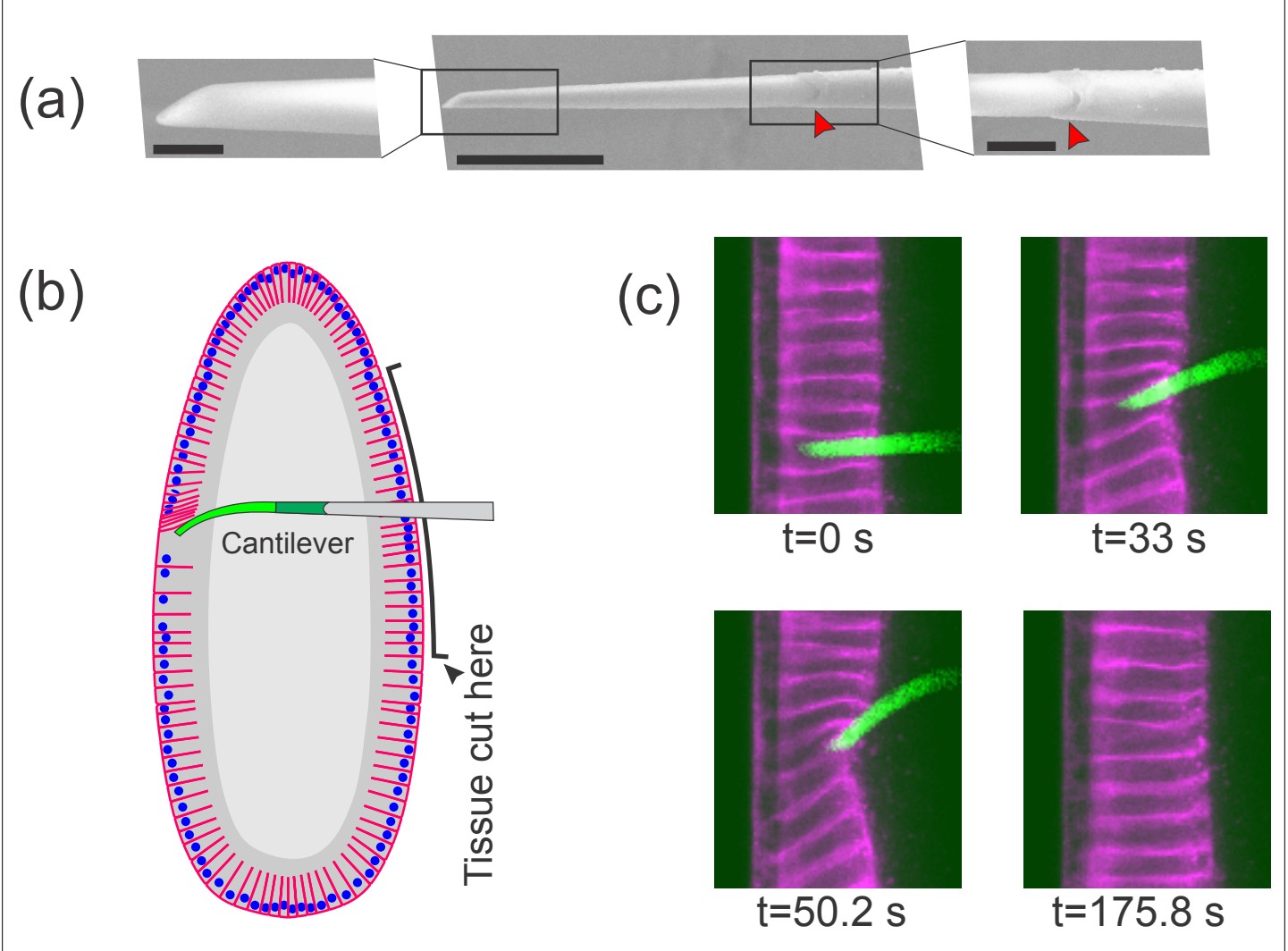

**Figure 1.** Flexible cantilevers can be used as a probe for tissue properties. (**a**) Scanning electron microscopy (SEM) image of a typical cantilever. Scale bars are 5, 30, and 10 µm respectively from left to right. Left and right panels are zoomed-in views of boxed regions in the central panel. The junction where the glass pipette ends and the flexible, polymer-only portion begins is indicated with a red arrowhead. Technical details of cantilever microfabrication are shown in *Figure 1—figure supplement 1*. (**b**) Schematic of the pulling experiment. The embryo is cut along the ventral side and the cantilever is inserted through the cut. At the relevant developmental stage, cells have a 'hole' on their basal side such that they are open to the interior. The flexible tip of the cantilever is inserted into a cell through the basal hole and translated so as to deform the cellular layer. (**c**) A sample measurement. To visualize cellular deformations arising from the applied force, cellular membranes are stained with a bright fluorescent dye (CellMask Deep Red) injected into the perivitelline space prior to measurement. The cantilever (green) is inserted into one of the cells (magenta) and is pulled upward. The resulting deflection of the cantilever as well as the deformation of the cellular layer is readily visible and quantifiable.

The online version of this article includes the following figure supplement(s) for figure 1:

**Figure supplement 1.** Details of cantilever microfabrication.

with the x-axis indicating time, and the y-axis displaying the position of the cell membranes (as measured from the fluorescent signal intersecting a curve drawn through the middle of the cellular layer). The kymograph for a representative pulling experiment is shown in *Figure 2a*.

Additionally, we represent the imaging data from each experiment as a deformation profile. For a given time point, we generate a curve showing the displacement of each membrane from its initial position as a function of its initial position (*Figure 2b*). Plotting several such curves, each representing a different time point of the experiment, together on one graph allows easy visualization of how tissue displacement spreads over time as the tissue is pulled. The deformation profiles for the loading and holding phases of a representative pulling experiment are shown in *Figure 2c and c''*, respectively.

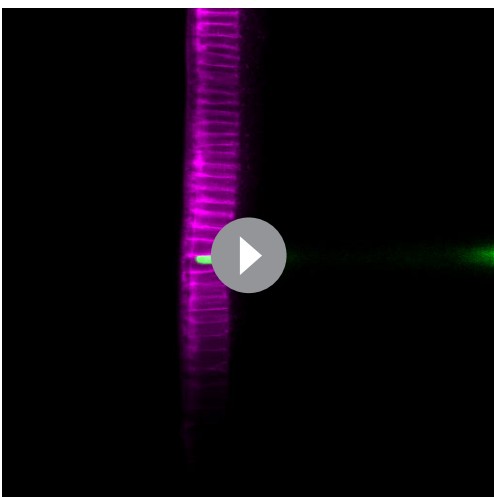

**Video 1.** Cantilever pulling experiment. Cell outlines are marked magenta (CellMask dye). Cantilever is green (boron-dipyrromethene [BODIPY] dye).
https://elifesciences.org/articles/85569/figures#video1

For some analyses, it is also useful to generate normalized deformation profiles, in which the displacement (or y) axis is linearly scaled so that the maximal displacement for each time point equals 1 (*Figure 2c'*). Importantly, measurements performed on different specimens are highly consistent (*Figure 2—figure supplement 1*).

One key advantage of the cantilever technique is that applied force can be calculated from cantilever deflection, which itself can be measured directly from the imaging data. This can be done because cantilever deflection is linearly proportional to the force being applied to it for the range of deflections observed. By Newton's third law, this force is equal in magnitude to the force the cantilever is exerting on the tissue.

To calculate the force, it is necessary to calibrate the probe. The force-deflection relation of each cantilever can be readily calculated using two pieces of information: the geometry of the individual cantilever, and the Young's (elastic) modulus of the PDMS material from which it is made. To calculate the Young's modulus of PDMS, we used three complementary techniques: (1) fabricating a macroscopic rod of PDMS to measure its deformation under its own weight (*Figure 3—figure supplement 1a-b*), (2) pulling the cantilever through a fluid of known viscosity to measure the resulting deflection (*Figure 3a–b*), and (3) probing the cantilever with an AFM probe (*Figure 3—figure supplement 1c-d*). The three methods estimate the Young's modulus of the PDMS comprising the cantilever to be 1.6 MPa, 0.9±0.17 MPa, and 2.1±0.23 MPa, respectively (the first value is based on a single measurement, see *Figure 3—figure supplement 1a-b*). The details of these measurements, along with the procedure we used to combine this information with individual cantilever geometry to calibrate each probe, are described in the Materials and methods section. Using this information, we calculated the force applied over the time course for five representative experiments (*Figure 3c*). The average force measured at maximal tissue displacement (at the end of the loading phase) was 11.1±1.2 nN, and fell to 5.0±0.45 nN by the end of the holding phase.

## Epithelial deformation during loading is nearly adiabatic

In order to correctly infer the various force contributions in a mechanical system, it is important to know whether the dynamics are adiabatic or not. An adiabatic process is defined as one in which no heat transfer takes place. Claiming that a process

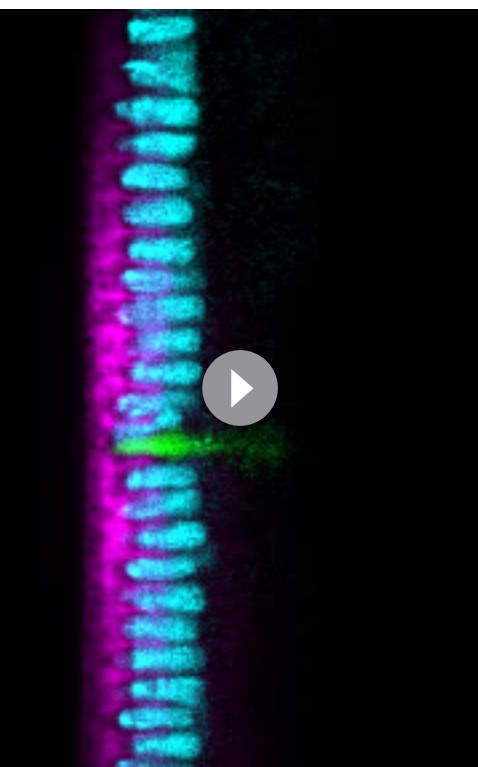

**Video 2.** Cantilever pulling experiment with nuclei. Cell outlines are marked magenta (CellMask dye). Cantilever is green (boron-dipyrromethene [BODIPY] dye). Nuclei (Histone-RFP) are cyan. Nuclei deform as the tissue is pulled and they restore their shape after the tissue is allowed to relax.
https://elifesciences.org/articles/85569/figures#video2

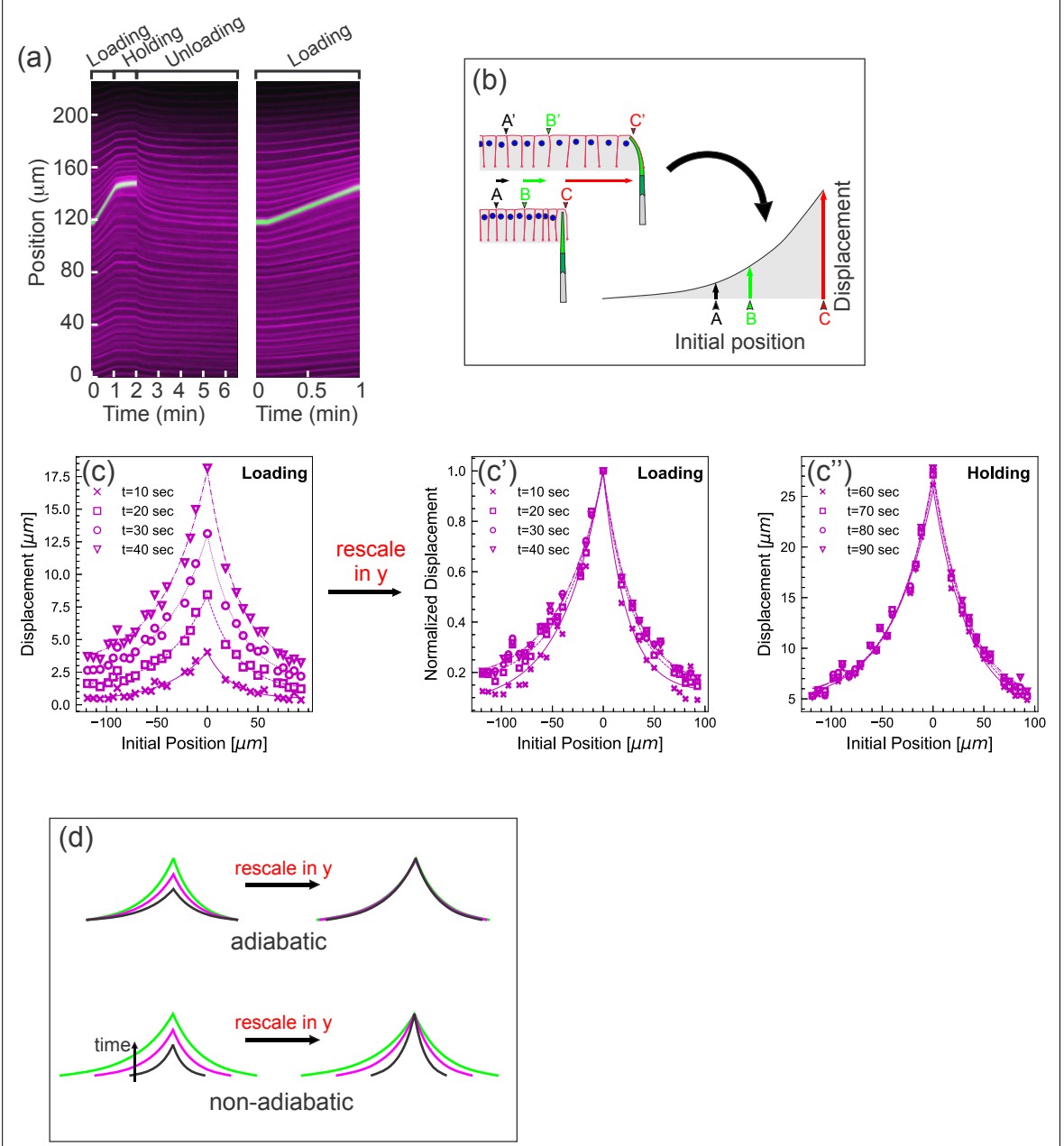

**Figure 2.** Quantifying tissue deformation during cantilever-based experiments. (**a**) Left kymograph corresponds to a complete loading-unloading cycle. Right panel is a zoomed-in portion of the same kymograph showing the loading phase only. Green streak reflects signal from the cantilever; magenta streaks reflect signal from the cell membranes. During the loading phase, when the cantilever base is being translated 0.5 μm/s, the green cantilever trace appears as a straight line, since the deflection of the soft cantilever tip is small relative to the translation of the base. (**b**) Schematic illustrating how a tissue deformation profile is constructed based on the data in a kymograph. (**c**) Tissue deformation profiles at several time points during the loading phase, extracted from the kymograph in (**a**). Deformation profiles were highly reproducible between individual experiment, shown in **Figure 2—figure supplement 1**. (**c'**) Same as (c), with the curves linearly re-scaled along the y-axis so as to have the same maximum at zero. (**c''**) Tissue deformation profile at several time points during the holding phase, also extracted from the kymograph in (**a**). (**d**) Cartoon showing the expected outcomes of linearly re-scaling tissue deformation profiles for force loading under two mechanical regimes. In adiabatic regimes, curves from different time points should appear roughly the same when linearly re-scaled along the y-axis only. In non-adiabatic regimes, linearly re-scaling along the y-axis only is insufficient to generate similar curves.

The online version of this article includes the following figure supplement(s) for figure 2:

**Figure supplement 1.** Tissue deformation profiles are highly reproducible between measurements.

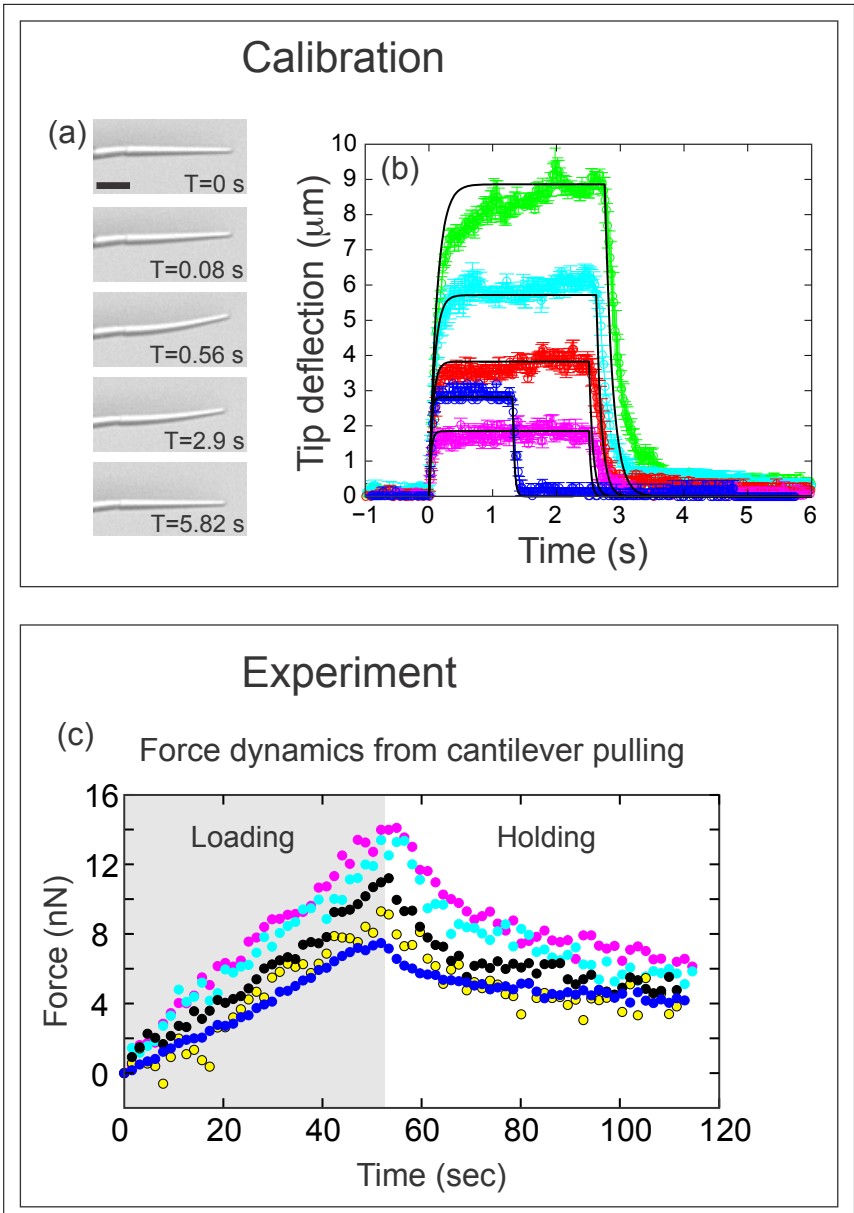

**Figure 3.** Quantifying applied force during cantilever-based experiments. (**a**) Images from force calibration experiment. A cantilever was dragged through a fluid of known viscosity (Tween 20). Different snapshots show the profile of the cantilever at different times. Scale bar: 20 μm. (**b**) Quantification from force calibration experiment. Cantilever tip deflection as a function of time for five different cantilevers being dragged through Tween 20. The flexible tip quickly reaches a steady-state degree of deflection when dragged at a fixed velocity, and thus is under the influence of a constant force. The approach used to calculate the elastic modulus of the boron-dipyrromethene (BODIPY)-infused polydimethylsiloxane (PDMS) from these data is given in the Materials and methods section. Additional approaches are described in *Figure 3—figure supplement 1*. The differences in magnitude of force deflection seen here are due to the different dimensions of each cantilever. Cantilevers used for in vivo experiments were individually calibrated taking the dimensions of the particular cantilever into account. (**c**) Force on cantilever as a function of time in five different in vivo pulling experiments. Force was calculated from cantilever deflection as described in Materials and methods. Only the loading (gray background) and holding (white background) phases of the experiments are shown, since the cantilever is no longer exerting force during the unloading phase. Maximal force at the end of loading was 11.1±1.2 nN; force at the end of the holding phase was 5.0±0.45 nN.

The online version of this article includes the following figure supplement(s) for figure 3:

**Figure supplement 1.** Complementary measurements used to determine the Young's modulus of the microfabricated cantilevers.

is adiabatic is equivalent to stating that friction is negligible, or alternatively, that the system is at equilibrium at every time point.

During the holding phase, most of the cell edges remain in a constant position, although the green cantilever trace and a few nearby cell edges move slightly upward (*Figure 2a and c''*). Since most motion ceases as soon as the cantilever stops moving, the tissue must have been largely in mechanical equilibrium at the time the cantilever stopped moving; in other words, most of the tissue was deforming adiabatically during the loading phase.

Further support for this conclusion comes from examining the loading phase more quantitatively. In previous work, we analytically and computationally explored the behavior of thin material sheets with a variety of mechanical properties (*Doubrovinski et al., 2017*). When these sheets were pulled at a single point, similarly generated deformation profiles from different time points of the same experiment were invariant up to re-scaling the y- and the x-axis by the factors $t^{\alpha}$ and $t^{\beta}$ respectively (see Figure S4 of *Doubrovinski et al., 2017*), where the values $\alpha$ and $\beta$ are determined by the mechanical properties of the system. In particular $\beta$ should exceed 0.5 if friction contributes significantly to the dynamics. In the pulling experiments presented here, we find that the deformation profiles obtained from the loading phase (*Figure 2c*) can be collapsed on almost the same master curve by re-scaling or normalizing the y-axis only; that is, $t^{\beta} = 1$, or equivalently $\beta = 0$ (*Figure 2c'*). This indicates that the tissue is deforming without significant input from external friction, which is consistent with the tissue deforming adiabatically (*Figure 2d*).

So far, we have demonstrated that the tissue is deforming largely adiabatically during the loading phase. During ventral furrow formation, the leading edge of the ectoderm travels toward the midline at a rate of approximately 0.1 µm/s (*He et al., 2014*). Since this is slower than the deformation rate in our experiments, dissipative forces such as friction must also be lower during ventral furrow formation. Therefore, we can conclude that tissue deformation is also largely adiabatic during this morphogenetic process.

## Epithelial deformation is dominated by elastic forces and boundary constraints

The time course of deformations and forces observed in our pulling experiments also reveal additional key mechanical features of the cellularizing epithelium.

One important feature is that the tissue is significantly elastic. During the loading phase (in which constant displacement is imposed), the applied force increases linearly with displacement, rather than with the rate of displacement (*Figure 3c*). If viscous forces were dominant then the force should be proportional to the rate of displacement, which is not the case. During the unloading phase, after the cantilever has been withdrawn, the tissue relaxes largely, though not entirely, back to its initial configuration (*Figure 2a*). These observations are both clearly consistent with the tissue itself being either an elastic or viscoelastic material.

The arguments given in the previous section for adiabatic deformation also mean that elastic forces dominate over friction and viscous shear forces in this system during the loading phase. If friction and viscous shear forces contribute only in a minor way during loading, what is primarily limiting tissue deformation in response to applied force? The observation that force on the cantilever drops by only about half (not to zero) during the holding phase suggests that boundary constraints may be important. To see why, consider a linear elastic spring which is immersed in a highly viscous fluid, and which is free at both ends. In response to force applied at one end, the trailing end will also move, though more slowly due to drag, and the spring will stretch. If the leading end is then held in a constant position, the trailing end will eventually catch up, the spring will eventually reach its resting length, and thus the spring will eventually cease applying a backward force on the leading end. If, however, this spring is fixed in place at its trailing end, a similar sequence of pulling then holding the free end will yield a different force profile. The spring will stretch while the free end is being pulled, but when the free end is then held fixed, the trailing end is unable to catch up. The spring should eventually reach a state of uniform deformation, but it cannot reach its resting length, and so it will continue to exert a force on whatever is holding the leading end in place. This analogy suggested to us that the reason the force on the cantilever never approaches zero during the holding phase of our pulling experiments is due to some type of boundary constraint. The nature of this 'boundary constraint' was not obvious to us at first, but will be explored in the computational studies below.

## Computation model of the *Drosophila* embryo

With these mechanistic insights in mind, we wanted to build a computational model both to test our preliminary conclusions and to estimate the elastic modulus of the cellular layer. To represent our experimental system computationally, we developed a three-dimensional (3D) model representing the entire embryo (*Figure 4*), in which geometries of all the relevant structures were taken from literature (*Mavrakis et al., 2008*) and measurements (*Figure 4b*). The cells are represented as open prisms with lateral and apical surfaces modeled as networks of elastic springs (each with spring constant *k*); the basal surfaces are left open (*Figure 4a''*). The cytoplasm filling the cell interiors and the yolk are modeled as a Newtonian fluid, with viscosity set to the experimentally measured value of 1000 cP (*Doubrovinski et al., 2017*; *Selvaggi et al., 2018*). The surface of the embryo is a rigid shell, with a no-slip condition imposed, representing the vitelline membrane, and the perivitelline fluid between the vitelline membrane and the cellular layer is approximated by a Newtonian fluid with the viscosity of water (~1 cP) (*Figure 4b*). External force is applied to a single cell edge, representing the effect of the cantilever.

Although the elastic components as described above are the springs, there is no obvious correspondence between individual springs and any particular cellular structure. In other words, a spring constant derived from this model would not reflect the mechanical properties of any particular cellular component. However, there is biological significance to the triangular network of springs which comprise a single cell surface; it behaves like an elastic shell or sheet, and is a good model for the biological cell surface (comprised of plasma membrane and its associated cytoskeletal network). Previous work (*Seung and Nelson, 1988*) has shown that in an elastic sheet modeled as a triangular network of springs, the elastic (or Young's) modulus of the sheet (*E*) and the spring constant (*k*) obey the following relationship: $E = 2k/\sqrt{3}$ . Therefore, elasticity will be represented by either *E* or *k* in the following discussion, depending on the context. Note that for this type of approximation, the Poisson's ratio is 1/3.

Technical details on the numerical implementation of our model are deferred to the Appendix. However, we briefly summarize key aspects of our modeling approach here. We used the immersed boundary method (*Peskin, 2002*; *Peskin and Printz, 1993*), which describes (1) how fluid responds to forces applied on it by an immersed solid object and (2) how the solid moves in response to external forces and fluid-solid interactions.

For the fluid domains (in this case, the cytoplasm and perivitelline fluid), we assume that the flow of the fluid is governed by Stokes equations. As a result, velocity of the fluid at any spatial location is uniquely determined from external forces driving the flow. In practice, to calculate this velocity distribution, we discretize the fluid domain by subdividing it into small tetrahedra with fluid velocities specified at every vertex (or node). The Stokes equations provide a relation between external forces and fluid velocities (*Happel and Brenner, 1973*). Thus, if we specify three spatial components of external force at *N* fluid nodes, we can obtain three components of velocities at each one of those *N* nodes by solving the discretized version of the Stokes equations.

The external force on the fluid comes entirely from the movement of the immersed solids, which this case consists of cell membranes. In this particular model, we need to calculate how the dynamics of the cell membranes are driven by the external force applied by the cantilever. As explained above, we model each cell membrane as a triangular network of springs. This network is in turn surrounded by fluid nodes, describing the fluid.

Each simulation step consists of five ordered sub-steps. (1) First, one calculates forces at the nodes of the triangulated elastic network. For most nodes, this is simply the sum of forces from elastic springs adjacent to that solid node; however, for nodes in contact with the cantilever, the force from the cantilever is also included. (2) Next, forces from the solid nodes are transferred to the fluid. The force on each fluid node is calculated from the force of the nearest solid nodes, with a correction for distance (see Appendix for details). (3) Now that external force on the fluid (due only to mechanical interactions with the immersed solid) is specified, velocities at fluid nodes may be calculated from the Stokes equations. (4) Next, velocities are transferred from fluid to solid. Specifically, the velocity on each solid node is calculated from the velocities of the nearest fluid nodes, with a correction for distance. (5) Next, coordinates of all solid nodes are updated by translating solid nodes with their now known velocities for a small interval of time *dt*. This modifies the configuration of the elastic solid and forces at all solid nodes must be recalculated anew. Iterating these steps, time evolution of the system is determined for some desired interval of time.

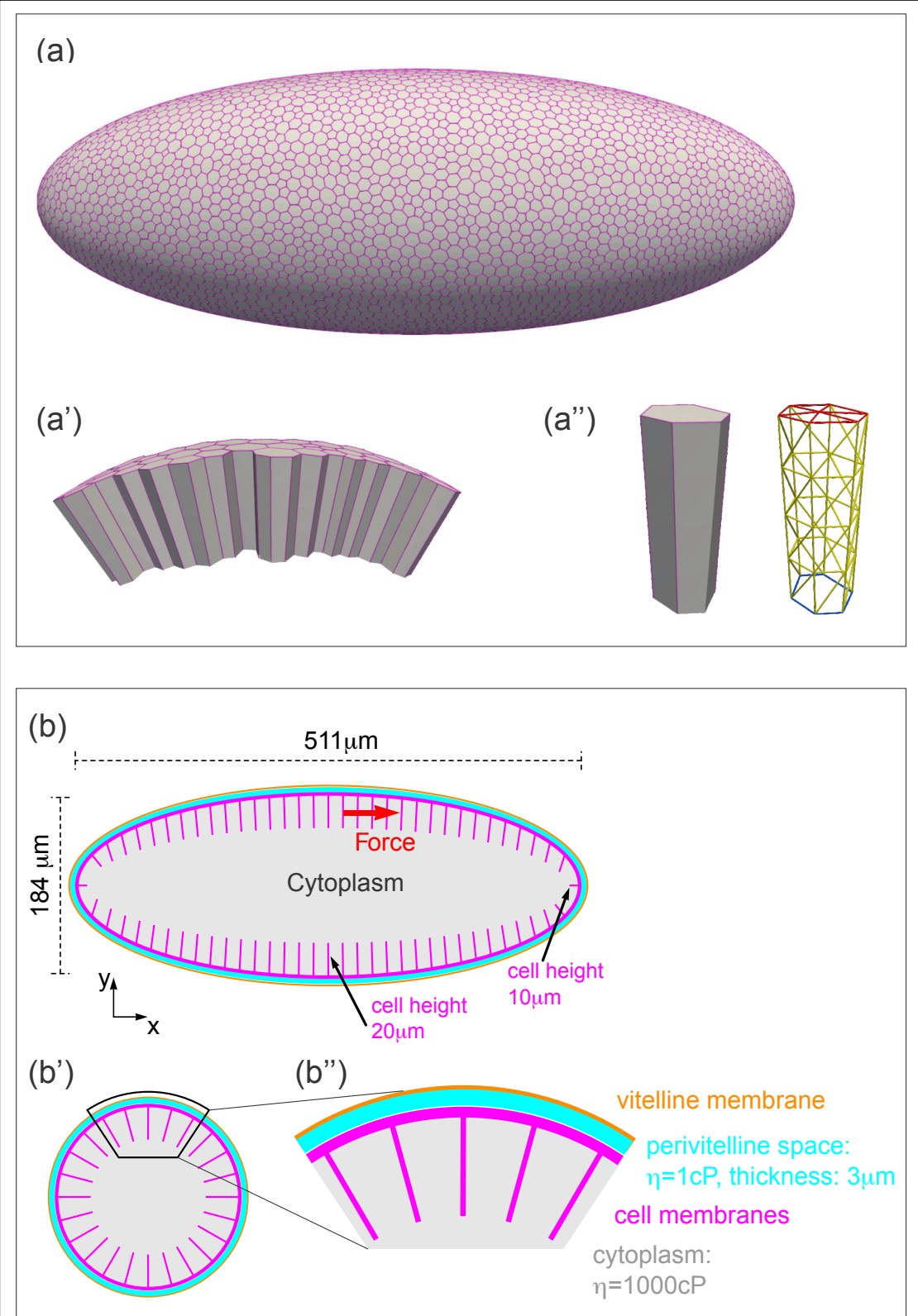

**Figure 4.** Three-dimensional computational model of the *Drosophila* embryo. (**a–a''**) Illustrations of the cellular structures represented in the computational model. (**a**) An ellipsoid was tiled with a disordered hexagonal network to represent the cell boundaries on the surface of the embryo. (**a'**) The cells themselves are three-dimensional prisms; a small portion of the surface, comprised of several cells, is shown here. (**a''**) Cells are represented as open prisms with lateral (yellow) and apical (red) surfaces modeled as networks of elastic springs, each with spring constant *k*; the basal

*Figure 4 continued on next page*

*Figure 4 continued*

(blue) surface of each cell is left open. (**b–b''**) Geometry of components of the simulated model embryo, not drawn to scale. (**b**) The long and short axis of the model embryo are 511 μm and 184 μm, respectively. Cell height is assigned in a gradient, with 20 μm heights in the middle diminishing to 10 μm heights at the poles. (**b'**) Cartoon cross-section of model embryo and (**b''**) a portion of the cross-section. The vitelline membrane is 3 μm thick and filled with a fluid of 1 cP viscosity. The fluid in the interior of the embryo is assigned a viscosity of 1000 cP, agreeing with published measurements (*Doubrovinski et al., 2017*; *Selvaggi et al., 2018*). Details of numerical implementation and validation of the numerical method are described in the Materials and methods, *Figure 4—figure supplement 1*, and *Figure 4—figure supplement 2*.

The online version of this article includes the following figure supplement(s) for figure 4:

**Figure supplement 1.** Schematic explaining terms used in the numerical model.

**Figure supplement 2.** Test case to benchmark numerical method.

## Assuming uniform mechanical properties fails to explain experimental data

During the unloading or relaxation phase, the computational model described above has only one free parameter: the spring constant, $k$, of the individual springs (or equivalently, the Young's modulus, $E$, of the cell surfaces). We could thus run the model with a range of values for $k$ and test which values, if any, fit the experimental data (*Figure 5*). To do this, we fixed a value for $k$ and imposed a

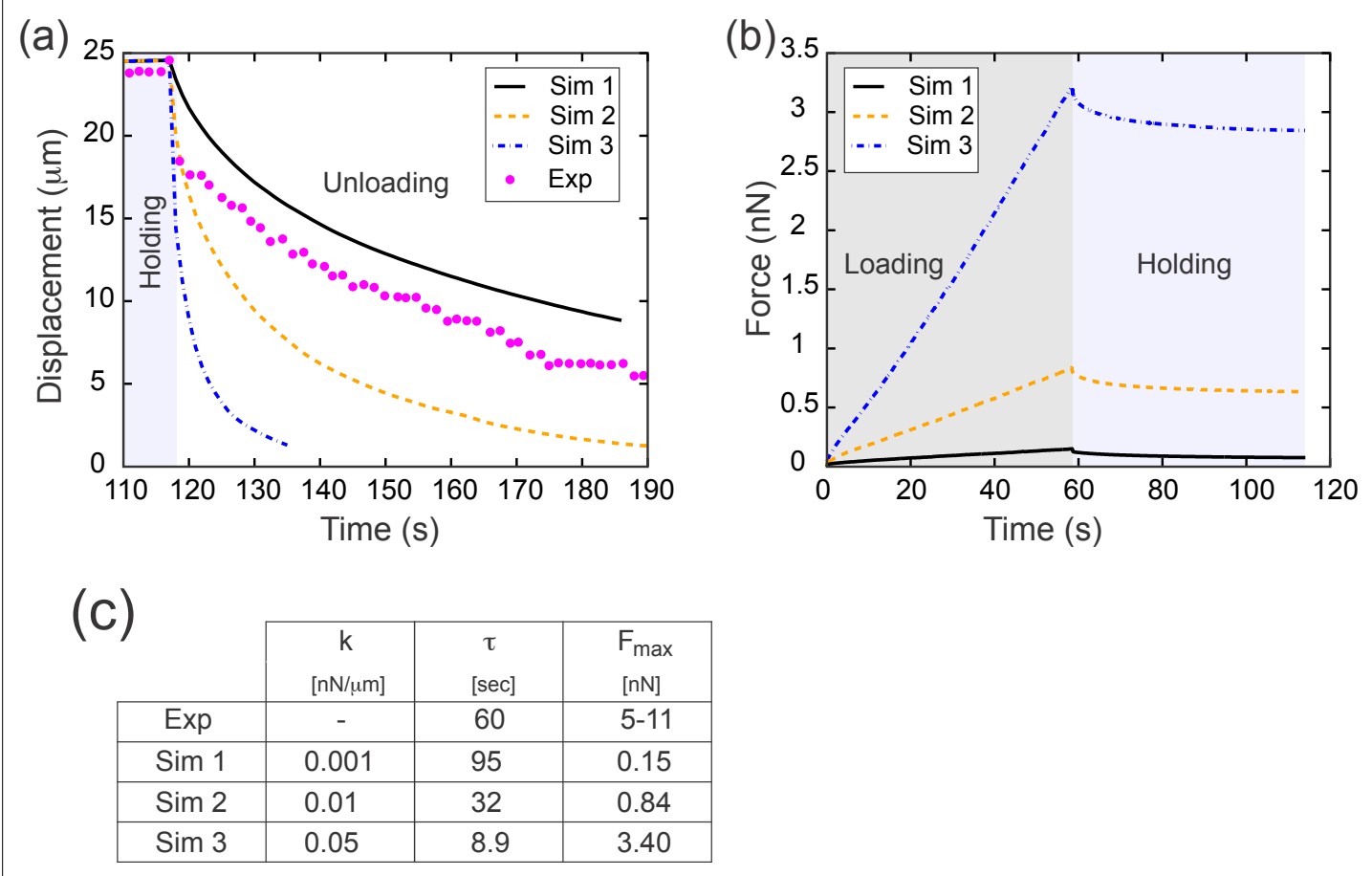

**Figure 5.** Results from parameter sweep with uniform elasticity explain relaxation dynamics or loading force but not both. (**a**) Parameter fitting for the spring constant $k$. The computational model was run with a range of values for the single free parameter $k$; see text. The externally applied force was removed to assess relaxation of the pulled edge during unloading. A constant value equal to the asymptotic value was subtracted from each curve; when plotted on a log plot, the slope of resulting line gives the exponential decay rate, $\tau$. (**b**) Force versus time for the model in (a) during the loading (gray background) and holding (light blue background) phases. Compare with *Figure 3c*, which shows the equivalent experimentally measured forces. (**c**) Table of decay rates and maximal force for experiment and model parameter sweep. The value of $k$ which best fits decay dynamics (0.001 nN/μm) drastically underestimates the loading force required.

| | k | τ | $F_{max}$ |
| --- | --- | --- | --- |
| | [nN/μm] | [sec] | [nN] |
| Exp | - | 60 | 5-11 |
| Sim 1 | 0.001 | 95 | 0.15 |
| Sim 2 | 0.01 | 32 | 0.84 |
| Sim 3 | 0.05 | 8.9 | 3.40 |

rate of tissue deformation at a single edge which matched that imposed experimentally during the loading and holding phases. We then removed the applied force and allowed the model tissue to relax (*Figure 5a*). To quantify the relaxation rate, we calculated an exponential decay rate $\tau$ from these data. Carrying out this parameter sweep, we found that a value of $k$=0.001 nN/μm (or equivalently $E$=0.0012) resulted in a curve that matched the data extremely well (*Figure 5a and c*, Simulation 1).

Having found that a spring constant of 0.001 nN/μm allowed us to realistically model experimental data from the unloading (or relaxation) phase of our experiments, we next wanted to test whether our model could also accurately capture the loading (or pulling) phase. During the loading phase, we had imposed a predetermined rate of displacement for a single edge, so by design the displacement matches that seen in vivo. However, in the loading phase we have an additional quantity that comes into play: the applied force needed to drive the tissue deformation. The approach we used to determine the instantaneous force needed in our model to generate the predetermined displacement is described in the Appendix. For each value of $k$ examined in our parameter sweep, we determined the force required for the tissue deformation and plotted this predicted force as function of time (*Figure 5b*). The predicted force increases linearly during the loading phase, which agrees well with the experimentally observed linear increase in force during loading (*Figure 3c*). However, for the value of $k$ which best matched the decay dynamics ($k$=0.001 nN/μm, Simulation 1), the predicted maximal force at the end of the loading phase was less than 0.2 nN (*Figure 5b and c*, Simulation 1). This value is an order of magnitude lower than the forces measured directly from the deflection of our calibrated cantilevers at the end of either the loading (~11 nN) or holding (~5 nN) phase. Conversely, for the value of $k$ (0.050 nN/μm, *Figure 5*, Simulation 3) for which the predicted maximal force (3.4 nN) most closely resembled the experimental data, the decay dynamics were far too rapid, with $\tau$=8.9 s, compared to $\tau$=60 s in the experimental data.

What is the reason for this discrepancy? Our predicted loading force was based on the assumption that the dominant effects on tissue mechanics (as captured in our model) are the same in the loading and unloading phase. Therefore, we began to consider ways in which this assumption might not be true. We thus began a series of computational experiments using simplified models to explore how specific features of the model affect loading and unloading.

## Floppy and stiff networks contribute differently to loading and unloading dynamics

One feature we examined was the effect of network topology on the dynamics. Recent work has explored the physical consequences of the so-called 'floppy' networks, in which the number of degrees of freedom exceeds the number of constraints, versus 'stiff' networks, in which constraints exceed degrees of freedom (*Broedersz and MacKintosh, 2014*; *Wyart et al., 2008*). For simplicity (and to reduce simulation time), we used a model tissue in which the epithelial layer was replaced by a simple network of elastic springs, and viscous forces from the cytoplasm and yolk were replaced by a constant drag force (see Materials and methods). We then explored the different effects of stiff versus floppy networks by using either a triangular (stiff) or hexagonal (floppy) network of springs.

During the initial loading phase, we imposed a fixed deformation rate at a chosen point, and determined the instantaneous external force required to achieve this deformation. We performed this simulation for models incorporating either triangular or hexagonal networks, and also varied the spring constant $k$ assigned to the springs or edges. For a given spring constant, the force required to deform the triangular network was greater than the force required to deform the hexagonal network by the same amount (*Figure 6a*). In spite of this quantitative difference, these networks responded to pulling in a similar way qualitatively: in both cases, the force needed for a given deformation increased as the elastic modulus increased (*Figure 6a*).

In the subsequent unloading phase, we removed the externally applied force and observed how the tissue relaxed over time (*Figure 6b*). In the triangular cases, the tissue relaxed back to its initial configuration more rapidly as $k$ increased (*Figure 6b and c*). This agrees with the intuition that tissue with higher elasticity should spring back more rapidly. In the hexagonal-mesh case, however, the tissue relaxed back to its initial position *more slowly* as $k$ increased (*Figure 6b*). Furthermore, the extent of tissue relaxation was also lower (*Figure 6b and d*). These results indicate that the response of hexagonal and floppy networks during unloading is qualitatively different.

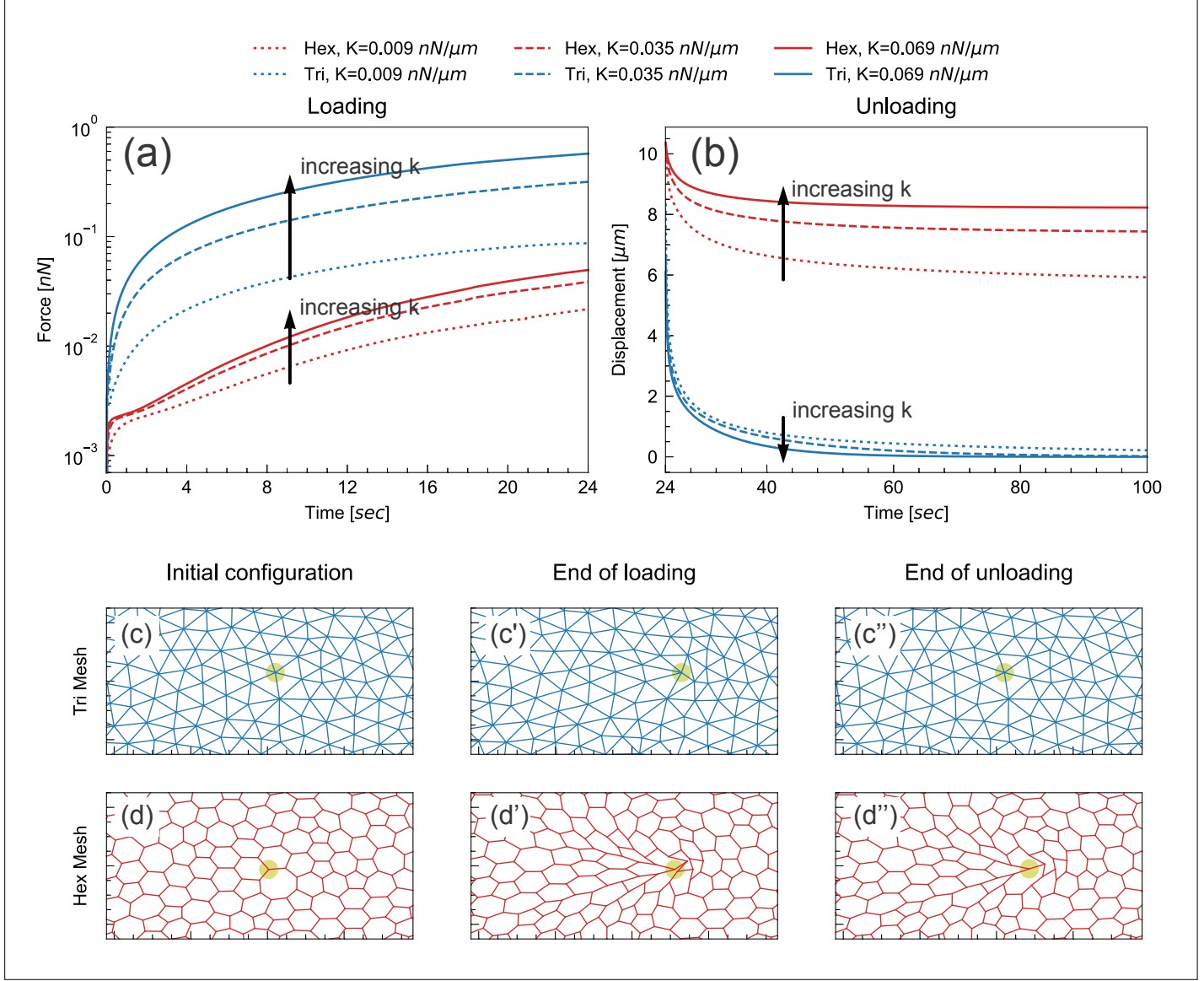

**Figure 6.** Floppy and stiff networks contribute differently to loading and unloading dynamics. (**a**) Time evolution of force during the loading phase in simplified model tissues with triangular or hexagonal networks of springs. For both triangular (blue) and hexagonal (red) networks, the force required to impose a fixed displacement increases with increasing spring constant $k$. (**b**) Time evolution of force during the unloading phase in the same model tissues. Tissue relaxation is faster with increasing $k$ in the triangular (blue) case, but slower with increasing $k$ in the hexagonal (red) case. (**c–d**) Time points from the same model, showing the part of the network surrounding the node where force is applied (indicated by pale green circle). Images shown are from simulations where $k$=0.035 nN/μm. The triangular network largely returns to its original configuration by the end of unloading (**c''**), but the hexagonal network remains significantly deformed at the same time point (**d''**).

## A difference in apical and basal elasticity determines tissue dynamics

There is an obvious parallel between these types of networks and different compartments of the cellularization-stage epithelia in our experiments. The apical surface appears most similar to a stiff elastic sheet, while the lateral surfaces and basal edges form a semi-hexagonal, and therefore floppy, network. Since the apical and basal-lateral compartments would therefore be expected to contribute differently to the loading versus unloading phases of our experiments, we hypothesized that our earlier mismatch of maximal forces calculated from the loading versus unloading phase may be due to a difference in the elastic modulus of the apical and basal-lateral compartments.

To test whether different compartments of the cellularization-stage epithelia exhibit different elastic properties, we modified an experimental setup we had used previously (***Doubrovinski et al.,***

*2017*). In those previous experiments, we injected a 20–30 μm diameter ferrofluid droplet into the embryo, positioned it in the cellular layer, and used a magnet to apply force across the basal-apical extent of the cell. In these previous experiments (reproduced in *Figure 7a–a'*), the speed of deformation decreased as the droplet approached the magnet, which we attributed to a build-up of elastic stress in the epithelium. In our new experiments, we used a smaller (approximately 10 μm diameter) ferrofluid droplet, which we could then position entirely inside the cell, allowing us to specifically apply force to a small region at the intersection of the apical and lateral surfaces of the cell, without directly contacting the basal edges (*Figure 7b*). In these experiments, the speed of deformation was constant or even increasing as the droplet approached the magnet, demonstrating that elastic stresses in this case were very small compared to the pulling force (*Figure 7b'*). This difference in dynamics demonstrates that the apical and lateral portions of the cell are softer, that is have a lower elastic modulus, than the basal edges.

We then returned to our full computational model of the embryo (*Figure 4*), this time assigning different values of the spring constant $k$ in different cellular compartments. To test the effect of varying elasticity in the floppy cellular components, we did a series of simulations in which we held $k_{apical}$ fixed and assigned increasing values to $k_{lateral}$ and $k_{basal}$ (*Figure 7c–c''*). Increasing $k_{lateral}$ and $k_{basal}$ 20- to 30-fold resulted in much higher values for maximal force $F_{max}$ (going from 0.2 nN to nearly 7 nN), but resulted in very similar values for the decay constant $\tau$ (~50 s). This validated our hypothesis that increased elasticity within floppy structural components should primarily affect mechanics during loading (characterized by $F_{max}$), with only minor effects on unloading (characterized by $\tau$). In contrast, when $k_{basal}$ and $k_{lateral}$ were kept approximately constant, but $k_{apical}$ was increased about sixfold, this resulted in a dramatic change of decay constant $\tau$, from 39 s to 9 s (*Figure 7—figure supplement 1*).

Notably, one of the parameter sets used in our simulations ($k_{apical}$ = 0.007, $k_{lateral}$ = 0.126, $k_{basal}$ = 6.825) resulted in a maximal force (7 nN) that fell within the 5 nN to 11 nN range of maximal force measured directly from cantilever deflection in experiments. Furthermore, despite the large values for basal/lateral elasticity (compared to $k$=0.05 in *Figure 5*), the fit to relaxation dynamics remained relatively good. To summarize, assuming that basal domains are much stiffer than the apical and the lateral domains suffices to reconcile the model with our measurement data. Note that this key assumption is independently verified in our ferrofluid experiments (*Figure 7a and b*) and that their interpretation in no way relies on either the cantilever-based experiments or their computational analysis. Note additionally that the final estimated values should be considered order-of-magnitude estimates, given the present measurement accuracy.

It is important to note that we expect that different distributions of elasticity among the floppy portions of the cell would yield similar results. Since we currently do not have experimental tools to measure, and thus computationally constrain, the ratio of lateral-to-basal elasticity, we cannot at this point precisely determine those numbers independently. However, our computational and experimental results combined do lead us to two firm conclusions. (1) The elastic modulus we estimated from relaxation dynamics ($E$ approximately = 0.001–0.01) provides a good estimate specifically of apical elasticity. (2) Basal elasticity in this tissue is substantially higher than apical elasticity.

## Tissue geometry imposes effective boundary constraints on in vivo dynamics

By examining our computational model, which closely reproduces the dynamics seen in vivo, we found ourselves able to answer a question that had puzzled us earlier. Why do tissue dynamics appear to be subject to boundary effects if the embryo is an ellipsoid and, thus, the cell layer appears not to have boundaries? One possibility is that some feature of the epithelium (.e.g., the topology of the cell network) stabilizes the tissue structure in a way that cells some distance from the cantilever would be immobile, forming a sort of pseudo-boundary. Plotting the displacement of cells in our model reveals that this explanation is not true. Even at short pulling times, cells throughout the model embryo move from their initial positions, even on the opposite side of the embryo from the force application, and even at the distant anterior and posterior poles (*Figure 8a–a'*, *Video 3*). We next examined previously acquired experimental data (*Doubrovinski et al., 2017*) to see if the same might be true in vivo. Indeed, when concentrated force was applied to one side of the embryo using a ferrofluid droplet, cells on the opposite side of the embryo clearly moved as well (*Figure 8b–b'*, *Video 4*). Although cells appear to be moving in a coherent direction (e.g., counterclockwise clockwise in *Figure 8b–b'*) along

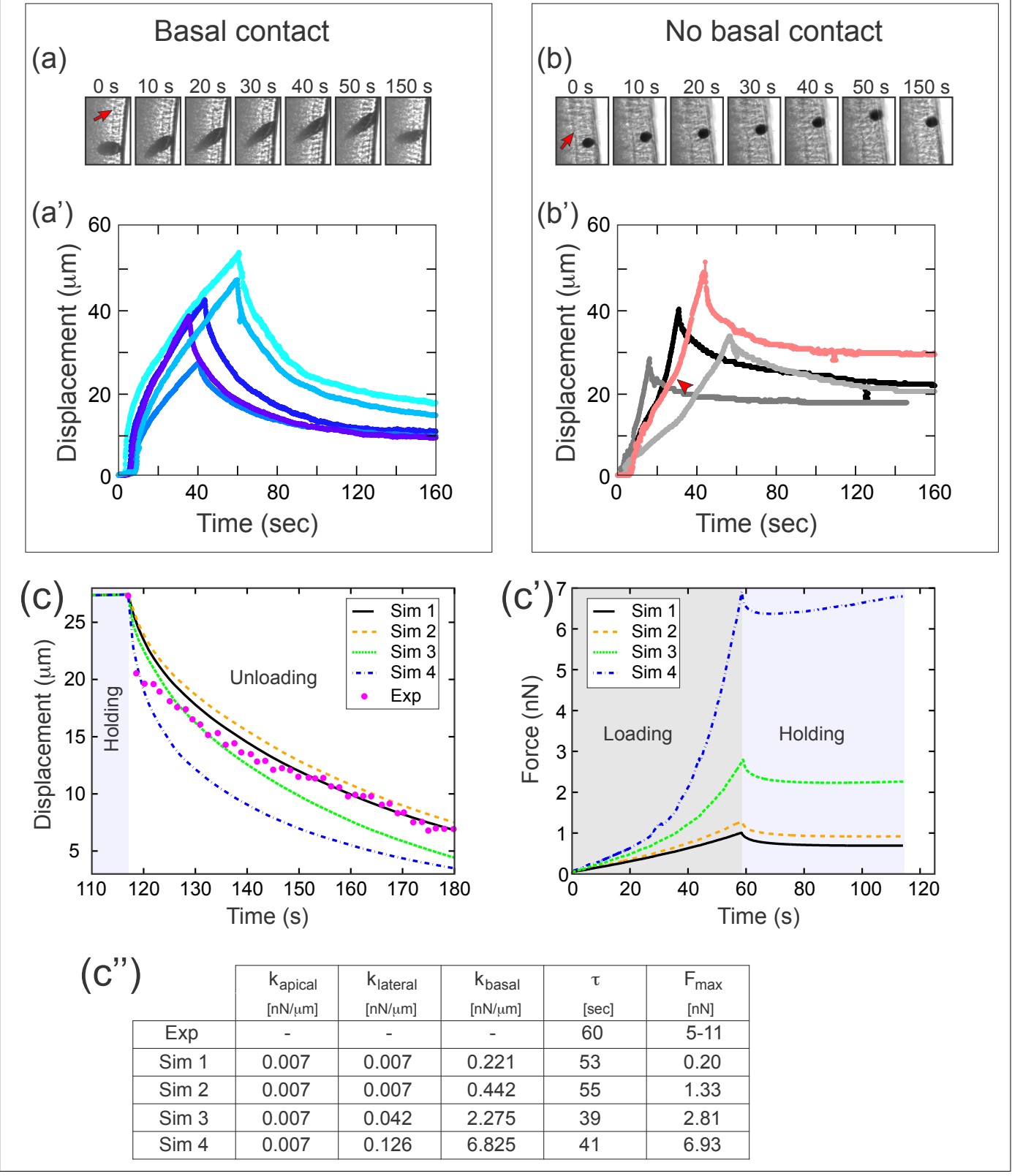

**Figure 7.** A difference in apical and basal elasticity determines tissue dynamics. (**a–a'**) Ferrofluid pulling experiment with droplets that contact the basal edge. The data in this panel has been previously reported (***Doubrovinski et al., 2017***). (**a**) Time frames from a typical experiment. The droplet was pulled toward the magnet and then released after approximately 50 s. The red arrow indicates the position of the cellularization front (i.e., the basal side of the newly forming cells). The droplet is clearly in contact with the basal side at all time points. (**a'**) Displacement of the ferrofluid droplet is plotted

*Figure 7 continued on next page*

*Figure 7 continued*

as a function of time for several experiments. For easiest comparison, the experiments chosen for this plot most closely match the total displacements in **b'**. The speed of these droplets decreased as they approached the magnet, indicating a build-up of elastic stress in the epithelium. (**b–b'**) Ferrofluid pulling experiment with smaller droplets that do not contact the basal edge. (**b**) Time frames from a typical experiment. The red arrow indicates the basal surface. In this sample, the droplet remains in the apical-lateral domain and does not contact the basal side at any time. (**b'**) Ferrofluid droplet displacement measured in four different pulling experiments. In the gray/black curves, the droplet is never in contact with the basal side. These droplets do not slow down as they approach the magnet, indicating that elastic stress is not building up. The pink curve corresponds to a droplet that made mechanical contact with the basal side at the beginning of the experiment, but moved apically and lost contact with the basal surface at the time point indicated by the arrowhead. The acceleration that occurs at this time point is further evidence that the basal domain is more strongly elastic than the apical-lateral domain. (**c–c''**) Parameter fitting to assess mechanical contributions of lateral and basal structures. The computational model was run with a range of values for the parameters $k_{lateral}$ and $k_{basal}$ while holding $k_{apical}$ fixed. Compare with *Figure 5* for fitting with a single spring constant $k$. Also see *Figure 7—figure supplement 1* for an additional simulation varying $k_{apical}$. (**c**) The externally applied force was removed to assess relaxation of the pulled edge during unloading. This data was used to calculate $\tau$. (**c'**) Force versus time for the model in (c) during the loading (gray background) and holding (light blue background) phases. (**c''**) Table of decay rates and maximal force for experiment and model parameter sweep. Increased elasticity within lateral and basal structures affects mechanics during loading (characterized by $F_{max}$), with only minor effects on unloading (characterized by $\tau$). Simulation 4 (blue curve) is our 'best fit', approximately matching both the rate of unloading and the maximal force seen in experiments.

The online version of this article includes the following figure supplement(s) for figure 7:

**Figure supplement 1.** The effects of apical elasticity on recoil.

the sagittal plane, the tissue is clearly not freely rotating, since the displacement drops off away from the applied force. A consideration of embryo shape suggests two obvious reasons why it does not rotate freely under applied force. Cells moving between the highly curved poles and the flatter lateral regions would need to change shape and/or neighbor relationships, and there may energetic barriers to this transition. In fact, our simulations reveal an upregulation of stress near the poles (*Figure 8c*). Additionally, if cells cannot rearrange completely freely, an imaginary ring composed of cells encircling the embryo crosswise must deform into a somewhat larger ellipse under the described tissue motion (*Figure 8d*), which would be expected to result in increased elastic tension among these cells. In summary, geometrical constraints preventing free rotation of the tissue allow for the build-up of elastic stress throughout the tissue when force is applied locally.

## Discussion

In this work, we have introduced a novel measurement technique that enables the application of concentrated force to a single epithelial cell in a living developing embryo. Crucially, this technique allows for the simultaneous quantification of the force being applied and the resulting tissue deformation. We envisage that our method, being simple and cost effective, will find applications in other systems.

Here, we used this technique to induce tissue deformations of a comparable rate and magnitude to the deformations that occur during *Drosophila* ventral furrow formation. This allowed us to directly assess the tissue material properties relevant for that morphogenetic process. During the loading phase of our pulling experiments, we found that epithelial deformation is adiabatic and that mechanics are dominated by elastic forces and geometric constraints. Since deformations in our experiments were comparable to (and even slightly faster than) the endogenous tissue deformation during ventral furrow formation, these conclusions should apply to that process as well.

To extract a quantitative measurement of tissue elasticity from our experiments, we developed a computational model. Surprisingly, our data could not be reconciled to a model in which the tissue was uniformly elastic along the apical-basal axis. In contrast, our data could be fit well by a model in which apical surfaces are significantly softer than basal surfaces. Strikingly, experiments using small ferrofluid droplets also showed that apical surfaces are significantly softer than basal surfaces. The conclusion that apical domains are softer might further be tested by studying embryos overexpressing folded gastrulation (fog), which upregulate the actomyosin cortex apically (*Dawes-Hoang et al., 2005*). Measurements in this background may allow us to directly test whether an increase in apical elasticity results in an increased rate of tissue recoil upon force removal, as predicted by our model.

Using a simplified computational model, we demonstrated that the reason apical and basal structures contribute differently to force loading and unloading is because the apical structures are topologically stiff, while the basal (and lateral) structures are topologically floppy. It is important to emphasize

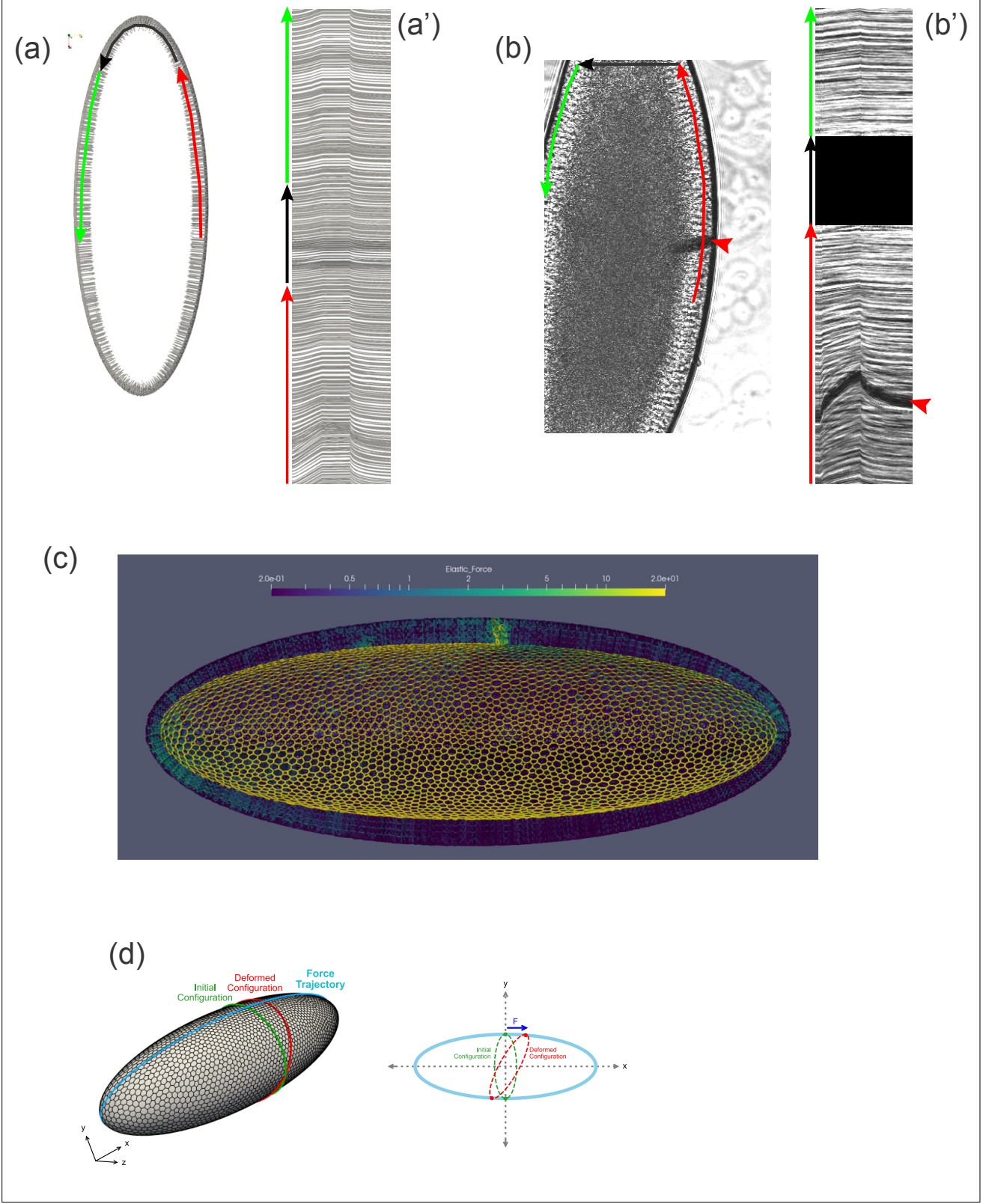

**Figure 8.** Tissue geometry constrains mechanical responses. (**a**) A thin section through the sagittal plane of the model from *Figure 7c* (Simulation 4), shown at a single time point. The red, black, and green arrows indicate the curves used to generate the kymograph in (a'). (**a'**) Kymograph corresponding to model data in (a). Traces at the top of the kymograph are not strictly horizontal, showing that motion is detected even on the opposite side of the model embryo from the force application. (**b**) A time point from a ferrofluid droplet pulling experiment. Ferrofluid droplet indicated by red

*Figure 8 continued on next page*

*Figure 8 continued*

arrowhead. The red, black, and green arrows indicate the curves used to generate the kymograph in (b'). (**b'**) Kymograph corresponding to experimental data in (b). Traces at the top of the kymograph are not strictly horizontal, showing that motion is detected even on the opposite side of the embryo from the droplet-applied force. (**c**) Color plot indicating the distribution of stress across the model embryo from *Figure 7c* (Simulation 4), averaged over the last 5 s of the pulling phase. Note that stress spreads across the whole tissue and is upregulated at the poles. (**d**) A schematic explaining how tissue geometry can affect the dynamics. An imaginary ellipsoid encircling the embryo is shown at two points: at the initial time point (green) and during force loading (red). Applying force at one point causes the imaginary ellipse to stretch, leading to an increase in elastic stress.

that, although the basal-open structure of the *Drosophila* embryo is unusual, the general principles described here should be universal. Specifically, relaxation dynamics in general should be expected to primarily reveal elasticities of stiff (non-floppy) components. This would apply even in systems in which cells are closed. For example, material properties of the apical and basal surface of a more typical epithelial monolayer would be expected to be reflected in in-plane relaxation dynamics, since they are topologically stiff. Material properties of the lateral surfaces, in contrast, would comprise a minor contribution.

Another unexpected result from this work is that geometrical constraints (which could alternatively be called system size effects because they depend on the system being 'small enough') generate a significant force opposing tissue movement when force is applied locally within the *Drosophila* embryo. As a result, any deformation, however small, propagates through the tissue on a time-scale of seconds. This effect was apparent in both our model and experimental data. Any model used to estimate the mechanical properties of the tissue must take this significant contribution to force balance into account. This was not realized in the previous related work on *Drosophila* embryo mechanics (*Bambardekar et al., 2015*; *D'Angelo et al., 2019*; *Doubrovinski et al., 2017*). Additionally, we envisage that similar physical effects may be equally relevant to experiments in other model systems such as zebrafish (*Behrndt et al., 2012*) and *Caenorhabditis elegans* (*Mayer et al., 2010*) and should thus be investigated.

It is of particular interest to compare our results with previous work on the system. In an important recent publication (*D'Angelo et al., 2019*), the authors performed magneto-rheology on cellularization-stage *Drosophila* embryos using 5 µm magnetic beads that were placed adjacent to the apical membrane. The authors carefully measured the response to a consistently applied magnetic force and found an abrupt developmental change in mechanical response, with tissue deformation becoming much greater beginning about 16 min before the onset of gastrulation. Based on fitting these data to a computational model in which a 2D viscoelastic sheet moving over a substrate represented the epithelium and vitelline membrane, the authors proposed that a sudden softening of the epithelium and an abrupt increase in friction with the vitelline membrane could account

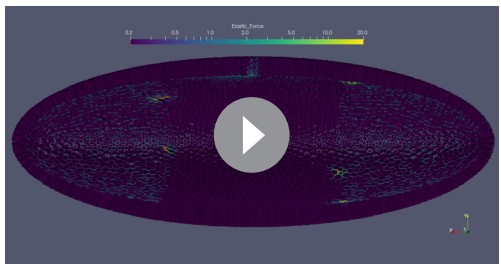

**Video 3.** Simulation of pulling experiment. The color indicates elastic stress distribution, calculated as $E*(L-L_0)$ for each elastic spring, where $E$ is the spring constant, $L_0$ is the rest length, and $L$ is instantaneous length. Note that stress is elevated (green) near the poles in the lateral membranes particularly during the holding phase.

https://elifesciences.org/articles/85569/figures#video3

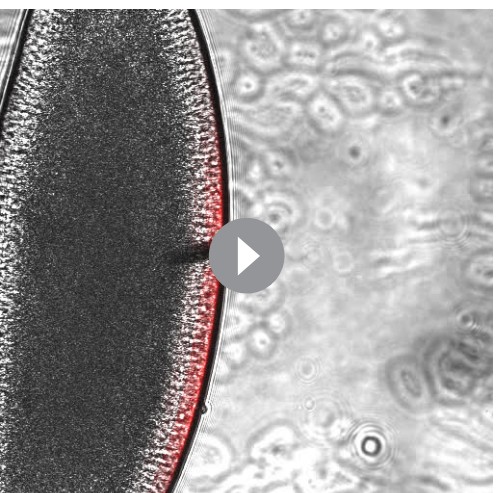

**Video 4.** Ferrofluid pulling experiment. Brightfield is shown in grayscale, CellMask dye fluorescence is shown in red. Note that the tissue is moving even on the opposite side of the embryo as the tissue is being pulled locally.

https://elifesciences.org/articles/85569/figures#video4

for their observations. Based on our work here, we propose an alternative interpretation. We note that during the early stage of cellularization, cells are short and the 5 µm apically localized beads would also be expected to be in mechanical contact with the very rigid basal side. At a later stage in cellularization, cells become taller and the bead should only contact the apical and/or the lateral cellular domains, which we have demonstrated are much softer. Furthermore, previous studies of cellularization dynamics are consistent with the basal edge of the cells moving past the 5 µm magnetic beads at approximately 16 min before gastrulation (*Figard et al., 2013*). We therefore propose that differences in the apparent material properties reported in *D'Angelo et al., 2019*, reflect differences in the material properties of the cellular domains being probed, and not time-dependent mechanical changes of the entire epithelial layer due to developmental progression. Importantly, our proposal also explains the observation that colcemid treatment blocks the apparent tissue softening (*D'Angelo et al., 2019*); microtubule depolymerization stops cellularization, so treated cells would remain short (*Royou et al., 2004*).

Additionally, our conclusion is consistent with the distribution of F-actin. Our previous data (*Doubrovinski et al., 2017*) showed that tissue elasticity is overwhelmingly dominated by cortical F-actin. At the same time, it is well established (*Sokac and Wieschaus, 2008*) that the basal side exhibits much stronger F-actin localization than either the apical or the lateral side. These data together strongly suggest that basal domains are much stiffer than both the lateral and the apical domains in complete accordance with our key conclusion.

In addition to qualitatively describing the relative mechanical contributions of different cell biological structures to the early *Drosophila* embryo, this work also resulted in a quantitative estimate for elasticity of the apical surface and a lower bound for elasticity of the basal domain. Combined with previous measurements of cytoplasmic viscosity, these measurements provide an essential set of physical parameters which are required for a truly predictive and quantitative model of morphogenesis in this system.

For a more complete model, future work should address two issues. One is the time-scale on which elastic energy in the system is stored. The fact that the force drops but then levels off during the holding phase implies two things: that some elastic stress dissipates on the time-scale of a minute, but that some elastic stress persists on very long time-scales. We previously established a lower bound for this rate (based on a combination of experiments and a highly simplified model) (*Doubrovinski et al., 2017*), but we did not establish a narrow range or even an upper bound. Future work should not only quantify this rate, but also address the underlying mechanism. One obvious candidate for a loss of stress would indeed be cell rearrangements, but we think this explanation is unlikely for two reasons. We see no evidence for cells intercalating, which would appear as cells disappearing or newly appearing in the kymographs. Also, if the cell topology changed, we would expect not just a change in the elasticity constant, but also in the 'resting length' - in other words, we would expect the tissue recoil to be incomplete. However, the tissue recoil is largely (85–100%) complete. The other obvious candidate for a loss of stress would be due to changes *within* each cell. Since elasticity depends on actin (*Doubrovinski et al., 2017*) and since actin turnover is fairly high in this tissue (*Figard et al., 2019*), we hypothesize that actin turnover is the underlying cause for the decreasing force seen during holding.

The other issue to address in the future is the relative elasticity of lateral versus basal domains. Our physically realistic model is too computationally intensive to expect to address this through parameter sweeps (see the 'Simplified model' section of Appendix for further discussion), so this issue should be addressed experimentally. One potential approach would be to examine mutants in which specific cellular domains have been perturbed. For example, the basal cellular domain in anillin mutants is noticeably compromised (*Field et al., 2005*), so measurements in this background could give further information on apical and lateral membrane properties.

## Materials and methods

**Key resources table**

| Reagent type (species) or resource | Designation | Source or reference | Identifiers | Additional information |
|---|---|---|---|---|
| Genetic reagent (*Drosophila melanogaster*) | gap43-Cherry | Gift from the lab of Eric F. Wieschaus | | https://www.nature.com/articles/nature07522 |
| Chemical compound, drug | Cell Mask Plasma Membrane Stain (Deep Red) | Thermo Fisher Scientific | C10046 | https://www.thermofisher.com/order/catalog/product/C10046 |

### Embryo collection and fl

All experiments used the membrane-Cherry fly line described in *Martin et al., 2010*. The fluorescent signal proved too faint for our purposes, so a membrane dye was used to visualize membrane outlines instead (see below).

Embryos were collected using on grape juice agar (Genesee Scientific) plates with a smear of yeast paste. To select embryos of a suitable stage, plates were covered in light halocarbon oil (Halocarbon Oil 27, Sigma). Embryos with a distinctive faint halo in the periphery, indicative of early cellularization, were transferred to small square mats of lab paper towel for further manipulations. The embryos were dechorionated in bleach, then quickly rinsed 10–12 times in distilled water. To attach the embryos to a slide, a thin streak of embryo glue (a saturated solution of adhesive from double-sided Scotch tape dissolved in heptane) was placed across the length of a coverslip and allowed to dry for 5–10 min. Dechorionated embryos were picked up and placed on top of the glue streak using an eyelash. A total of 8–12 embryos were chosen for each round of imaging. The embryos were then desiccated in a closed container with Drierite (Fisher Scientific) for 22 min before further manipulations.

### Microinjection of fluorescent dye

Needles used for dye microinjections were fabricated from borosilicate capillaries (B100-75-10, Sutter Instruments) using a pipette puller (P-2000, Sutter Instruments). Pipettes were backfilled with 25-fold diluted CellMask Deep Red Plasma Membrane stain (Thermo Fisher Scientific) using a gel loading tip (Eppendorf). Pipettes were broken at the tip by gently nudging the pointed end against an edge of a coverslip, under a drop of oil.

To inject dye, the tip of a pipette was carefully pushed against the vitelline membrane so as to insert it into the perivitelline space without piercing the cellular layer. An Eppendorf injector (FemtoJet 5247, Eppendorf) was used with the following settings: compensation pressure: 10 hPa, injection pressure: 220 hPa. Typically, the injection was performed using the 'clean' command such that maximal pressure was applied.

After injection, the embryos were cut along the side opposite to the site of injection for subsequent cantilever insertion.

### Fabrication of cantilevers

A glass pipette was pulled the same way as done when making microinjection needles. The pipette was then similarly fractured at the tip by gentle contact with a glass coverslip, but in air with no added oil. To make the polymer filling for the glass needle, PDMS (Sylgard 184, Dow Corning) was mixed with a stock solution of BODIPY (BODIPY 493/503, Thermo Fisher Scientific; 1 mg per ml stock solution in anhydrous DMSO) in proportions 3:2 (by volume). The mixture was placed in a dish of aluminum foil and left overnight at 80°C to evaporate off DMSO. Next, the PDMS-BODIPY mixture was mixed with PDMS crosslinker in proportions 10:1 by weight. This mixture cures within 48 hr if left at room temperature, but may be stored for months at –20°C without curing. A small droplet of PDMS-BODIPY-crosslinker mixture was placed in the center of a weighting boat. A glass pipette is placed into the weighting boat so as to immerse its (open) tip in the droplet, simultaneously leaning the other end of the pipette against the edge of the weighting boat. The weighting boat is left at 80°C for 5 hr allowing PDMS to cure. Finally, the pipette is carefully lifted out from a (now cured) PDMS droplet. In this way, the inner part of the pipette tip is filled with crosslinked fluorescent PDMS.

The PDMS-filled pipette was bevelled using a micropipette grinder (E-45, Narishige) for 20 s with the speed dial set to 7. This is needed to remove the very end of the tip since otherwise the very tip of the cantilever will be too thin and will tend to curl on itself.

Finally, the outer layer of glass was etched away using Armour Etch (Armour Glass Etching Cream, 2.8 Oz), a paste comprising a mixture of hydrofluoric acid and fluorites. A spectrophotometer cuvette was filled with Armour Etch cream halfway, and a layer of light silicone oil (Silicone oil for oil baths, Sigma) was added on top to prevent solvent evaporation. The tip of the pipette was immersed into the etching paste and was left to etch for 3–5 min. To observe the pipette tip through a vertically mounted microscope, a retrofitted MicroForge (MF-900) with a custom mount for the cuvette was used (*Figure 1—figure supplement 1d*). This completely dissolved the exposed glass layer to expose the core of cured PDMS. The pipette was then lifted out from the cuvette and gently washed with a solution of Triton X-100 and saline (0.4% NaCl and 0.03% TX-100 in distilled water). Finally, the pipette is bent 90 degrees in flame of a cigarette lighter to simplify its mounting in a pipette holder.

It is important to note that filling capillaries from the front, by capillary action, is absolutely critical for generating cantilevers of reproducible properties. When microfabricating cantilevers using pulled glass pipettes as a mold, one may try introducing PDMS into the pipette by backfilling (as we in fact did initially). When microcantilevers are fabricated in this way, their Young's modulus (measured by subjecting the probe to viscous drag) varied widely. To track the origin of this undesired variability, we performed focused ion beam (FIB) milling on our microfabricated probes. In the course of milling, half of the cantilever beam was milled away along its length so as to expose and examine the mid-cross-section of a sample using scanning electron microscopy. It was found that portions of the cantilever were in effect hollow, due to sub-micron-sized bubbles trapped in the polymer (*Figure 1—figure supplement 1a*). We believe that bubbles were formed due to the air trapped inside the capillary tip when it is being backfilled. In contrast, when capillaries are filled from the front using capillary action, FIB milling reveals no bubbles trapped within the polymer (*Figure 1—figure supplement 1b*), and measurements of the Young's modulus are consistent.

## Micromanipulation protocol and microscopy

The glass pipette with its PDMS cantilever tip was translated using a piezo actuator (PXY200SG two-axis piezo actuator, Newport). The piezo was controlled with an amplifier (NPC3SG three-channel piezo amplifier, strain-gauge position control, Newport) interfaced with MATLAB. The MATLAB code used to control the piezo is available for downloaded from https://github.com/doubrovinskilab/cantilever_embryo_rheology (copy archived at *Cheikh, 2023*). To mount the piezo actuator on a microscope stage, an optical breadboard was screwed to the stage using two custom threaded holes. A three-axis stage (PT3 - 1" XYZ Translation Stage with Standard Micrometers, Thorlabs) was mounted on the breadboard. The piezo actuator was attached to the three-axis manual stage using two 45-degree angle brackets and a custom adopter mount.

Confocal laser scanning microscopy was done using a Zeiss LSM 700 with excitation wavelengths of 488 nm (green) and 639 nm (far red).

## Computational methods

The computational methods used in this paper (including cantilever calibration, models, and numerical implementation) are described in detail in the Appendix.

## Acknowledgements

We thank Jeffrey Woodruff for helpful discussions. We also thank Salena Fessehaye for discussion of the manuscript. We gratefully acknowledge the help of the Characterization Center for Materials and Biology at UT Arlington, in particular Dr. Jiechao Jiang and Prof. Efstathios I Meletis, for their help with AFM characterization of the probe. We thank the Nano3 facility at UCSD for their help with the cryo-EM aspects of the project. We thank Saiana Khandarkhaeva, Natalia Dubrovinskaia, and Leonid Dubrovinsky for their help with the FIB modification of the commercial AFM probes. The work was supported by the Robert A Welch Foundation (grant I-1950-20180324) and the NIH (grants 1R01GM134207 and 1R21HD105189).

## Additional information

### Funding

| Funder | Grant reference number | Author |
|---|---|---|
| National Institute of General Medical Sciences | 1R01GM134207 | Mohamad Ibrahim Cheikh<br>Joel Tchoufag<br>Miriam Osterfield<br>Swayamdipta Bhaduri<br>Konstantin Doubrovinski |
| Eunice Kennedy Shriver National Institute of Child Health and Human Development | 1R21HD105189 | Mohamad Ibrahim Cheikh<br>Joel Tchoufag<br>Miriam Osterfield<br>Swayamdipta Bhaduri<br>Konstantin Doubrovinski |
| Robert A. Welch Foundation | I-1950-20180324 | Mohamad Ibrahim Cheikh<br>Joel Tchoufag<br>Miriam Osterfield<br>Swayamdipta Bhaduri<br>Konstantin Doubrovinski |

The funders had no role in study design, data collection and interpretation, or the decision to submit the work for publication.

### Author contributions

Mohamad Ibrahim Cheikh, Conceptualization, Software, Formal analysis, Investigation, Methodology, Writing - original draft; Joel Tchoufag, Conceptualization, Software, Methodology; Miriam Osterfield, Conceptualization, Supervision, Funding acquisition, Investigation, Methodology, Writing - original draft, Writing - review and editing; Kevin Dean, Software; Swayamdipta Bhaduri, Investigation, Writing - original draft; Chuzhong Zhang, Investigation; Kranthi Kiran Mandadapu, Conceptualization; Konstantin Doubrovinski, Conceptualization, Formal analysis, Funding acquisition, Investigation, Methodology, Writing - original draft

### Author ORCIDs

Miriam Osterfield http://orcid.org/0000-0002-8907-852X
Kevin Dean https://orcid.org/0000-0003-0839-2320
Konstantin Doubrovinski http://orcid.org/0000-0002-4403-948X

### Decision letter and Author response

Decision letter https://doi.org/10.7554/eLife.85569.sa1
Author response https://doi.org/10.7554/eLife.85569.sa2

## Additional files

### Supplementary files

• MDAR checklist

### Data availability

All simulation code used in the study is publicly available under https://github.com/doubrovinskilab/cantilever_embryo_rheology, (copy archived at *Cheikh, 2023*).

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

## Appendix 1

### Computational analysis of cantilever calibration

In order to obtain the Young's modulus of a cantilever, the cantilever was dragged at a constant speed of 53 μm/s through Tween 20, a fluid whose density is density $\rho = 10^3 \text{kg/m}^3$ and viscosity $\mu_f = 460$ cP (information provided by Sigma-Aldrich). Using these data and the measured dimensions of the cantilevers calibrated, the characteristic Reynolds number can be estimated to $Re \sim 10^{-5}$, implying that inertial effects are negligible. The dynamics of the cantilever is thus governed by the balance between viscous and elastic forces. Viscous drag was calculated based on the slender body theory following *Cox, 1970*, which is justified by the small diameter-to-length ratio of the cantilever. In particular, the drag force per unit length reads:

$$F_v\left(x,t\right) = 2\pi\mu_f \frac{\partial h}{\partial t} \left( -\frac{2}{\ln\left(2R_a\right)/L} - \frac{1}{\ln\left(2R_a/L\right)^2} \left[ 1 + 2\ln 2 + \ln\left(1 - \left(\frac{2x}{L}\right)^2\right) - \ln\left(\frac{R\left(x\right)}{R_a}\right)^2 \right] \right),$$

where $R\left(x\right) = R_{min} + x\left(R_{max} - R_{min}\right)/L$ and $R_a = \left(R_{max} + R_{min}\right)/2$ are the local and average radii of the tapered cantilever, respectively, $h$ is vertical deflection at a given position ($x$) and time ($t$), and $L$ is the length of the cantilever. On the other hand, since the cantilever length remains constant, we use inextensible beam theory to obtain the expression for the force density due to bending (*Semler et al., 1994*):

$$F_e\left(x,t\right) = -\frac{\partial^2}{\partial x^2} \left( EI\left(x\right) \frac{\partial^2 h}{\partial x^2} \frac{1}{\left[1 + \left(\partial_x h\right)^2\right]^{3/2}} \right),$$

where $I\left(x\right)$ is the area moment of inertia (at position $x$). To determine the value of the Young's modulus $E$, we performed a parameter sweep varying the value of $E$. To do this, we solve the equations following from the force balance (along the direction perpendicular to the axis of the cantilever and assuming small displacement): $F_v + F_e = 0$. This was done using the open solver *FreeFEM* (*Hecht, 2012*).

### Numerical model

Here, we describe the numerical method used to simulate tissue deformation. Our approach is a combination of the immersed boundary method (IBM) (*Cuvelier et al., 1986*; *Peskin, 1972*; *Peskin and Printz, 1993*) and the finite element method (FEM) (*Boffi et al., 2007*). The chosen methodology is motivated by the requirement to describe fluid-structure interactions when treating the dynamics of embryonic cells that are elastic on the surface and filled with viscous fluid in the interior (*Doubrovinski et al., 2017*).

The IBM approach involves considering an elastic solid immersed in a fluid as shown in *Figure 4—figure supplement 1*. In what follows, $\Omega_f$ denotes the region occupied by the fluid with boundary $\Gamma_f$, while $\Gamma_s$ is the interface of the fluid-immersed elastic solid. Initially, the fluid domain $\Omega_f$ is discretized into finite elements (*Figure 4—figure supplement 1b-d*). Within the IBM approach, the geometry of this (Eulerian) grid remains fixed throughout a simulation. The interface of the solid is meshed into triangles (whose edges will be modeled as linear springs, see *Figure 4—figure supplement 1b*). The nodes of the corresponding discretized (Lagrangian) solid will be translated according to the relevant equations of motion which we turn to next.

### Governing equations

Describing the evolution of the solid interface (the cell surfaces) requires simulating the ambient fluid, whose dynamics are given by the Stokes equations:

$$\nabla \cdot u_f = 0, \tag{1}$$

$$\mu \nabla^2 u_f - \nabla p_f = f_f. \tag{2}$$

Here, $\mu$ is dynamic viscosity, while $u_f\left(x,t\right)$ and $p_f\left(x,t\right)$ are the velocity and the pressure fields, respectively. Force density acting on the fluid is $f_f$. In our case, $f_f$ derives entirely from forces

originating from the immersed structure $\Gamma_s$ . We are justified in neglecting inertia since a typical Reynolds number in our problem may be estimated to be on the order of $Re=10^{-8}$ . More specifically, in a typical experiment, embryonic tissue is subjected to concentrated force as the tissue is pulled by a cantilever. The typical speed $U$ with which the cantilever is translated was $0.005 \, \mu m/ms$. The typical size (length) of an embryo $L$ is $500 \, \mu m$. The viscosity of the embryonic cytoplasm was previously measured to be on the order of $1.0 \, nN \cdot ms/m^2$ (*Doubrovinski et al., 2017*; *Selvaggi et al., 2018*). The density of the cytoplasm is on the order of that of water $\rho = 10^{-6} g/m^3$ . Hence, we obtain $Re = \frac{\rho UL}{\mu} = 2.5 \times 10^{-8}$ as above.

To describe the surfaces of model cells, we assume that the cortex may be described as a thin linearly elastic shell characterized by a (2D) Young's modulus and a Poisson's ratio.

Finally, the dynamics of the solid are coupled to that of the fluid by imposing a no-slip boundary condition, such that the velocity of the fluid matches that of the solid at the solid interface. The net force density on the solid ($f_s$ , which is defined below) is equated with the force on the fluid (through the $f_f$ term). Note that these are the key standard assumptions exploited in a typical IBM implementation (*Peskin, 1972*; *Peskin and Printz, 1993*).

## Numerical implementation

For the fluid, we start by casting the governing equations for the fluid into their weak form:

$$\int_{\Omega_f} \left( \nabla \cdot u_f \right) \psi d\Omega_f = 0, \tag{3}$$

$$\mu \left( \int_{\Omega_f} \nabla u_f \cdot \nabla \phi d\Omega_f - \int_{\Gamma_f} \partial_n u_f \phi d\Gamma_f \right) - \left( \int_{\Omega_f} p_f \nabla \phi d\Omega_f - \int_{\Gamma_f} p_f \phi n d\Gamma_f \right) = \int_{\Omega_f} f_f \phi d\Omega_f. \tag{4}$$

where $\psi$ and $\phi$ are test functions for the pressure and the velocity field, respectively (*Boffi et al., 2007*; *Boffi and Gastaldi, 2003*; *Cuvelier et al., 1986*). The spatial domain is discretized into P1b-P1 finite elements which satisfy the inf-sup condition (*Boffi and Gastaldi, 2003*; *Boffi et al., 2007*; *Cuvelier et al., 1986*). More specifically, we choose the cubic P1b element for the velocity test function $\phi$ and the linear P1 element for the pressure test function $\psi$. Note that there are $N$ discrete nodes for the velocity field and $M$ nodes for the pressure field in what follows.

Using the above discretization, the relevant fields are approximated as:

$$u_f = \sum_{j=1}^{N} \widetilde{u}_{fj} \phi_j \left( x \right), \quad p_k = \sum_{k=1}^{M} \widetilde{p}_{fk} \psi_k \left( x \right). \tag{5}$$

In terms of the discretized variables the Stokes equations yield the Galerkin form:

$$\sum_{j}^{N} \widetilde{u}_{fj} \int_{\Omega_f} \left( \nabla \cdot \phi_j \right) \psi_i d\Omega_f = 0, \quad \forall i = 1, \dots, M \tag{6}$$

$$\mu \sum_{j}^{N} \widetilde{u}_{fj} \left( \int_{\Omega_f} \nabla \phi_j \cdot \nabla \phi_i d\Omega_f - \int_{\Gamma_f} \partial_n \phi_j \phi_i d\Gamma_f \right) - \sum_{k}^{M} \widetilde{p}_{fk} \left( \int_{\Omega_f} \psi_k \nabla \phi_i d\Omega_f - \int_{\Gamma_f} \psi_k \phi_i n d\Gamma_f \right)$$
$$= \int_{\Omega_f} f_f \phi d\Omega_f, \quad \forall i = 1, \dots, N. \tag{7}$$

*Equations 6 and 7* define a finite element system of $3N+M$ equations and $3N+M$ unknowns (in the 3D case). Assuming that the fluid boundary is Dirichlet, the linear system will take the standard form

$$\begin{bmatrix} \mathbb{A} & \mathbb{B} \\ \mathbb{B}^T & \mathbb{O} \end{bmatrix} \begin{bmatrix} \widetilde{u}_f \\ \widetilde{p}_f \end{bmatrix} = \begin{bmatrix} F \\ 0 \end{bmatrix} \tag{8}$$

where

$$\mathbb{A} = \mu \int_{\Omega_f} \nabla \phi_j \cdot \nabla \phi_i d\Omega_f, \tag{9}$$

$$\mathbb{B} = \int_{\Omega_f} \psi_k \nabla \phi_i d\Omega_f, \tag{10}$$

$$\boldsymbol{F} = \int_{\Omega_f} f_f \phi_j d\Omega_f. \tag{11}$$

The plasma membrane surfaces of the embryo can be thought of as a set of interconnected linearly elastic 2D plates. A particularly time-efficient implementation for representing elastic plates involves discretizing each plate as a set of interconnected triangles whose edges are linear springs (*Seung and Nelson, 1988*). More specifically, it may be shown that in the limit of small deformation, the discrete triangulated mesh approximates a 3D linearly elastic material whose Young's modulus is given by $E = 2k/\sqrt{3}$ (with $k$ being the spring constant of the mesh) and Poisson's ratio $\sigma = 1/3$ (*Seung and Nelson, 1988*). Therefore, our immersed solid $\Gamma_s$ is modeled as a triangular mesh of springs. The no-slip boundary condition at the fluid-solid interface amounts to moving the solid nodes with the local velocity of the fluid:

$$\frac{dX_s}{dt} = u\left(X_s, t\right).$$

where $X_s$ is a given point on the surface of the solid. Every spring comprising the discretized solid contributes a force dipole at its two adjacent nodes:

$$F_s = k\Delta L \hat{s},$$

where $F_s$ is the magnitude of the elastic force at a solid node, $k$ is the spring constant, $\Delta L$ is the difference of the corresponding instantaneous spring length and its rest length, while $s$ is a unit vector along the respective spring.

The coupling of solid and fluid dynamics is implemented within the IBM framework in a standard way (*Peskin, 1972*; *Peskin and Printz, 1993*). In brief, force density of the solid is transferred to the adjacent fluid nodes (the 'spreading' step), contributing the $f_f$ -term in *Equation 2*. Next, the velocity and pressure fields of the fluid are obtained from the system of linear equations (*Equation 8*). Finally, in the 'interpolation' step, the velocity of the fluid is interpolated to the solid nodes and those nodes are then advanced through a time-step using a Euler-forward scheme.

## Validation of the numerical scheme

In this section, we validate our numerical solver by comparing simulation results to the solution of an analytically tractable fluid-structure interaction problem. The problem, illustrated in *Figure 4—figure supplement 2*, involves stretching an elastic plate submerged in the middle of a viscous fluid. The plate is fixed on the left, with the right end subjected to a constant force.

Neglecting inertia, we can write the equations governing the stretching of the plate as:

$$-2\mu \frac{\partial}{\partial y} u(x, y, t)|_{y=H} = E \frac{\partial^2 \delta(x, t)}{\partial x^2} \qquad \forall x \epsilon \left[0, L\right] \& t > 0, \tag{12}$$

$$\delta(0, t) = 0 \qquad \forall t > 0, \tag{13}$$

$$E \frac{\partial \delta(x, t)}{\partial x}\bigg|_{x=L} = F \qquad \forall t > 0, \tag{14}$$

$$\frac{\partial \delta(x, t)}{\partial x}\bigg|_{t=0} = 0 \qquad \forall x \epsilon [0, L] \tag{15}$$

$$\delta(x, 0) = 0 \qquad \forall x \epsilon \left[0, L\right] \tag{16}$$

where $\delta(x,t)$ is the displacement of the plate, $E$ the 2D Young's modulus of elasticity, $\mu$ the viscosity of the fluid, and $\frac{\partial \delta(x,t)}{\partial t} = u(x, H, t)$ is the $x$-velocity component of the fluid. *Equations 13 and 14* are the boundary conditions for the problem, *Equations 15 and 16* are the initial conditions, and *Equation 12* represents the balance of forces acting on the plate (equating stress in the fluid to force density on the solid). The factor of 2 in *Equation 12* comes from the fact that viscous drag is present on both sides of the elastic plate.

To solve this system analytically, we assume the distance between the plate and fluid boundaries is small in comparison to the length of the plate (i.e., $H \ll L$). In this case, the flow is well approximated by the Poiseuille solution such that the $y$-velocity is identically zero, whereas $x$-velocity $u(x, H, t)$ is linear in between the elastic plate and the horizontal walls of the fluid domain. With these assumptions, *Equation 12* may be recast as a diffusion problem

$$\eta \frac{\partial \delta(x,t)}{\partial t} = E \frac{\partial^2 \delta(x,t)}{\partial x^2} \qquad \forall x \epsilon\, [0, L] \ \& \ t > 0, \tag{17}$$

where $\eta = 2\mu/H$ and $D = E/\eta$ is the diffusion coefficient. The solution of this problem is given by the Fourier expansion

$$\delta(x,t) = \frac{f}{E}x + \frac{2f}{LE} \sum_{n=1}^{\infty} \frac{(-1)^n}{\beta_n^2} exp\left( \frac{-\beta_n^2 E}{\eta} t \right) sin(\beta_n^2 x), \qquad \text{where } \beta_n = \frac{2n-1}{2L}\pi, n \epsilon N_{>0} \tag{18}$$

which may be considered a sum of the steady-state solution and a transient. This expression, which is the analytical solution to the problem, is shown as the red line in *Figure 4—figure supplement 2b-c*.

We also simulated the problem using our numerical scheme to obtain an approximate solution. This simulated solution is shown as the blue line in *Figure 4—figure supplement 2b-c*.

The analytical expression and numerical simulation are in very close agreement. Moreover, as expected, the agreement systematically improves with decreasing thickness of the fluid layer (data not shown). Taken together, simulation results in *Figure 4—figure supplement 2* provide a rigorous validation of the chosen numerical scheme.

## Numerical simulations of experimental data

In this section we detail simulations used to interpret the pulling experiments probing the rheology of embryonic epithelium. The geometry of a model *Drosophila* embryo is illustrated in *Figure 4*. The computational domain is an ellipsoid whose dimensions approximately match the well-documented dimensions of an embryo (*Mavrakis et al., 2008*). The outer solid boundary represents the rigid vitelline membrane. Beneath the vitelline membrane is a thin (3 μm) gap of perivitelline fluid. The rest of embryonic interior is described as a uniform Newtonian fluid with a viscosity of 1000 cP, as previously measured (*Doubrovinski et al., 2017*; *Selvaggi et al., 2018*). Cells are hexagonal prisms whose faces are elastic plates modeled by a set of inter-connected springs (*Figure 4a–a''*). Dimensions of cells are based on experimental data.

A P1b-P1 FEM mesh was generated so as to discretize the whole computational domain (including the model perivitelline space, cytoplasm, and yolk). The mesh was further refined in the region corresponding to the perivitelline fluid. Dirichelet (no-slip) condition was prescribed at the outer domain boundary corresponding to the vitelline membrane.

To capture the cantilever pulling experiments, simulation is divided into three phases:

1. In the **loading phase**, one cell edge is subjected to force a pulling force for 60 s. In our experiments, the imposed speed of the cell edge that was being pulled remained approximately constant during the course of loading. To match the measurements, in our simulations, the external pulling force was applied through a proportional control system (P-controller) such that the externally applied force is given by

$$\overrightarrow{F}_{external}(t) = K_p \overrightarrow{e}(t) - \overrightarrow{F}_{internal}(t) \tag{19}$$

$K_p$ was set to be 0.2 nN/μm; $\overrightarrow{F}_{internal}$ represents the sum of the internal forces on the cell edge being pulled; the error value $\overrightarrow{e}(t)$ is the difference between the measured and the simulated displacement of the cell edge that is being pulled. The external force (e.g., *Figure 7c'*) in our simulations was obtained from *Equation 19*.

2. In the **holding phase**, the cell edge was held fixed for an additional time of 60 s.
3. In the **unloading phase**, the external pulling force was removed.

All simulation code is publicly available under https://github.com/doubrovinskilab/cantilever_embryo_rheology (copy archived at *Cheikh, 2023*).

## Simplified model

To better understand simulation results obtained from the model in the previous section, we considered the behavior of floppy and stiff networks in the experimentally relevant geometry (*Figure 6*). The full model is extremely computationally intensive; for example, Simulation 4 in *Figure 7c'* required 3 weeks of simulation time using 200 processor cores. Therefore, we used a simplified model to allow us to more quickly explore parameter space. One simplification was including 2D spring networks confined exclusively to the surface of the ellipsoid representing the surface of the embryo (using a soft force of constraint). The more significant simplification in terms of computational time was that we replaced explicitly modeled Stokes fluid with effective drag.

As discussed below, the results from the simplified treatment cannot be quantitatively compared to the results from the full treatment including hydrodynamic interactions. However, to only gain qualitative insights into the behavior of our system, it is admissible to replace Stokes fluid dynamics with viscous drag, especially considering that we are mainly interested in the behavior of the solid and not the fluid. In our simulations (*Figure 6*), the networks were confined exclusively to the surface of the ellipsoid representing the surface of the embryo using a soft force of constraint. We checked that increasing the magnitude of the force of constraint by an order of magnitude did not affect the dynamics. All other aspects of these simulations were the same as in the case of the full hydrodynamic model.

For an example of how viscous drag can provide an inaccurate approximate for the physics modeled by Stokes fluid dynamics, consider the following example. A thin 2D disc moving through a 3D ambient Stokes fluid experiences force that is proportional to the radius; in contrast, using standard assumptions for drag, the force would be proportional to the area of the disc (*Happel and Brenner, 1973*). The use of drag in this example only approximates the results obtained from explicitly modeling the fluid in the limit where the fluid on either side of the disc is thin and bounded by another solid surface nearby (*Happel and Brenner, 1973*). Since the embryo is filled with a fully 3D fluid, we therefore chose the more physically correct model for our main simulations in spite of the limits this placed on our ability to efficiently explore large regions of parameter space.

