## [Editor Report]

Using a novel micropipette-based, minimally invasive approach in combination with theoretical and computational analysis, this important work probes tissue mechanics in the *Drosophila* embryo. The authors provide compelling evidence for the applicability of their method, which reveals important differences between the mechanical properties on the apical and basal tissue sides. This work should be of broad interest to scientists studying tissue mechanics, membranes, and developmental processes.

---

## [Decision Letter]

**Decision letter after peer review:**

Thank you for submitting your article "A comprehensive model of *Drosophila* epithelium reveals the role of embryo geometry and cell topology in mechanical responses" for consideration by *eLife*. Your article has been reviewed by 2 peer reviewers, one of whom is a member of our Board of Reviewing Editors, and the evaluation has been overseen by Michael Eisen as the Senior Editor. We apologize for the lengthy delay in furnishing this decision letter.

The reviewers have discussed their reviews with one another, and the Reviewing Editor has drafted this to help you prepare a revised submission. As you can see, the reviewers are very supportive of your work, but they do ask for clarification on some key points.

Essential revisions:

1) While the theoretical/computational sections of the paper are well developed, some of the heuristic arguments about elasticity and dissipation could be fleshed out in more detail. Particularly in the case of the discussion around springs embedded in a viscous medium, with and without distant constraints. These details should be shown in the main body of the paper, not in the supplement, in a form easily understandable by most readers.

2) Have the authors investigated any potential developmental mutants?

Or can they at least suggest which known mutants would be the most appropriate to test out their ideas?

3) During such large deformations, the nuclei of the cells should rearrange to compensate for cell shape changes. How could this affect the observed force-displacement in the experiment? During the unloading period, do nuclei return to their original position?

4) Likely, these experiments are laborious and have low throughput, but is it possible to perform the same measurements when the cellularization has not progressed much, for example, when the lateral membranes are half-through? This could be interesting to test against the theoretical model.

5) It seems that the same setup can be used to impose apical-basal displacement by inserting the cantilever into the embryo but waiting until the cellularization is finished and the basal holes of the cell are closed. Considering the difficulty of these experiments, having such measurements can greatly help 3D modeling of the tissue at this stage and later stages, which has been extensively studied.

6) The analogy at the top of page 10 explaining the residual force in the holding phase is not evident to us. Imagine we have a 1D chain of springs (initially all stretched) in a viscous fluid with the two ends of the chain pinned. If we exert an external force on a node, all springs will deform until the system reaches a force balance. In my opinion, this is a better representation of the tissue than a single free spring. How could this explain the observed force reduction during holding? Also, how do the authors reject the possibility of cellular rearrangement during the holding phase, which could result in lower force during the holding?

7) Several conclusions are derived based on a comparison between experimental measurements and a rather simplified mechanical model of the 3D epithelium. For example, one major conclusion is that the apical surface of the cells is softer than the basal side, which is very important for understanding morphogenesis and 3D tissue dynamics. How robust are these conclusions to model choice? For example, if one includes more details in the model, such as impermeable membranes that limit the rate of cell volume change, does the mechanical asymmetry between apical and basal sides still stands? Are there other studies that found similar results? Is it possible to back up these conclusions with more experiments, for example, in a mutant background with stiffer apical junctions, and show that their model can reproduce the relaxation timescales?

7) The authors reported measurements without error bars or units in several places throughout the manuscript. For example, on page 7, there should be error bars for the cantilever Young modulus measurements or E on top of page 12 or k in line 350. The authors should go through these issues and report appropriate numbers.

Similarly, the name *Drosophila* should be in italics throughout the text (it is mixed now).

---

## [Author Response]

Essential revisions:1) While the theoretical/computational sections of the paper are well developed, some of the heuristic arguments about elasticity and dissipation could be fleshed out in more detail. Particularly in the case of the discussion around springs embedded in a viscous medium, with and without distant constraints. These details should be shown in the main body of the paper, not in the supplement, in a form easily understandable by most readers.

We understand that many readers may not be familiar with the Immersed Boundary Method and may therefore find our simulations hard to interpret. We thank the reviewers for pointing this out. For this reason, we have now included a short non-technical account of the Immersed Boundary Method in the end of the section entitled “Computational Model of the *Drosophila* Embryo” (lines 257-288).

2) Have the authors investigated any potential developmental mutants?Or can they at least suggest which known mutants would be the most appropriate to test out their ideas?

We have not yet investigated any developmental mutants using our technique, though we certainly plan to pursue such measurements in the future. As suggested, there are mutants which could help test some of our hypotheses. In particular, anillin mutants (in which basal rings are compromised) could be used to test whether the basal surface is indeed more elastic than the apical and lateral surfaces, while folded gastrulation (fog) overexpression could be used to test whether apical elasticity is indeed the major determinant of tissue recoil. We have now added a discussion of these points to the relevant paragraphs in the Discussion section (lines 529-532 and 448-452).

3) During such large deformations, the nuclei of the cells should rearrange to compensate for cell shape changes. How could this affect the observed force-displacement in the experiment? During the unloading period, do nuclei return to their original position?

To address this comment, we have now performed additional experiments on embryos whose nuclei are fluorescently labeled with histone-RFP. An additional supplemental movie has been included as supplemental Video 2. It is seen that nuclei do deform as the tissue is being pulled and that they restore their shape completely after the tissue is allowed to relax (lines 975-977).

Although these results themselves don't answer whether nuclei make a significant contribution to elasticity, we have previously published results that indicate that elasticity is overwhelmingly determined by the actin cortex, and therefore not by internal structures such as nuclei. We failed to mention this in the first version of this manuscript, but we have corrected this here (lines 69-70). Thank you very much for bringing this issue to our attention.

4) Likely, these experiments are laborious and have low throughput, but is it possible to perform the same measurements when the cellularization has not progressed much, for example, when the lateral membranes are half-through? This could be interesting to test against the theoretical model.

All of our experiments were in fact done at the stage when the lateral membranes were approximately half formed (that is, between 15 and 20 microns long), though we now realize that we didn't describe this in detail in our manuscript. We have now added a note to this effect in the Figure 2 – supplement 1 legend (lines 914-916). In order to ensure reproducibility (and to simplify the modeling), we aimed to perform measurements at as similar a stage of cellularization as possible, though we do understand your point that analyzing data at a variety of membrane lengths could be a valuable complementary approach.

5) It seems that the same setup can be used to impose apical-basal displacement by inserting the cantilever into the embryo but waiting until the cellularization is finished and the basal holes of the cell are closed. Considering the difficulty of these experiments, having such measurements can greatly help 3D modeling of the tissue at this stage and later stages, which has been extensively studied.

Yes, we believe that our setup could indeed be used to impose apico-basal deformations after the basal holes close. However, imaging such experiments would be very challenging using our current confocal microscope setup. This is because cells are between 30 and 40 microns tall at the time when basal holes close, so visualizing the relevant deformations would require imaging very deep inside the embryo. Unfortunately, this is very challenging using our current confocal microscope setup, though we anticipate that this would be fairly straightforward using two-photon imaging. Since we currently don’t have access to a two-photon setup which we can modify to mount the necessary measurement equipment on the microscope stage, we have not been able to perform such measurements. It would certainly be interesting to pursue this approach in the future.

6) The analogy at the top of page 10 explaining the residual force in the holding phase is not evident to us. Imagine we have a 1D chain of springs (initially all stretched) in a viscous fluid with the two ends of the chain pinned. If we exert an external force on a node, all springs will deform until the system reaches a force balance. In my opinion, this is a better representation of the tissue than a single free spring. How could this explain the observed force reduction during holding? Also, how do the authors reject the possibility of cellular rearrangement during the holding phase, which could result in lower force during the holding?

The analogy at the top of page 10 (to a free spring in viscous fluid) seemed the most intuitive to us at first, since there are no obvious points at which the tissue is fixed in place. However, the point of that paragraph is in fact to argue that our data *doesn't* behave like a free spring, but in fact does behave more like a spring with pinned ends (i.e. fixed boundaries), which agrees with your intuition.

However, neither type of spring (free or pinned) would be expected to exhibit force reduction during the holding stage *if the pulling was done adiabatically*. (Specifically, for a free spring, the force would be zero during holding, while for a pinned spring the holding force would be the same as the very end of the pulling phase.) That the pulling was in fact done adiabatically is confirmed by analyzing the deformation profiles during pulling (discussed in lines 183-195) and by observing that cell motion stops abruptly when pulling stops (discussed in lines 178-182).

The resolution to the apparent paradox is simple in theory; adiabatic pulling and a loss of force during holding can be reconciled if the elastic stress dissipates when the tissue is held in a deformed state – in other words, if the material is not actually an ideal spring.

One obvious candidate for a loss of stress would indeed be cell rearrangements. We think this explanation is unlikely for two reasons. (1) We see no evidence for cells intercalating in our data. In kymographs, cell intercalation events would appear as cells disappearing or newly appearing in the plane of view, and we really don't see this. (2) If the cell topology changed, we would expect not just a change in the elasticity constant, but also in the "resting length" – in other words, we would expect the tissue recoil to be incomplete. However, the tissue recoil is largely (85 – 100%) complete.

The other obvious candidate for a loss of stress would be due to changes *within* each cell. Since elasticity depends on actin (Doubrovinski et al. 2017) and since actin turnover is fairly high in this tissue (Figard et al. 2019), we hypothesize that actin turnover is the underlying cause for the decreasing force seen during holding.

We realize now that we didn't properly discuss the drop in holding force in the first version of the manuscript. We have now added a more thorough discussion of this issue (lines 510-525). Thank you for raising this point.

7) Several conclusions are derived based on a comparison between experimental measurements and a rather simplified mechanical model of the 3D epithelium. For example, one major conclusion is that the apical surface of the cells is softer than the basal side, which is very important for understanding morphogenesis and 3D tissue dynamics. How robust are these conclusions to model choice? For example, if one includes more details in the model, such as impermeable membranes that limit the rate of cell volume change, does the mechanical asymmetry between apical and basal sides still stands? Are there other studies that found similar results? Is it possible to back up these conclusions with more experiments, for example, in a mutant background with stiffer apical junctions, and show that their model can reproduce the relaxation timescales?

In our view, the most direct conceivable test of our key prediction (that different cellular domains have different stiffness) is to apply force to different cellular domains individually and study their resulting mechanical response. This is what we do in Figures 7a,b, using ferrofluid droplets of different sizes. There we show that when a ferrofluid droplet is used to stretch the cellular layer, the elastic response is only seen if the droplet makes contact with the basal side of the epithelium. This result is completely independent from and in no way relies on either the cantilever-based measurements or the computational model used to analyze those measurements. In order to emphasize this point, we added a comment to this effect on lines 388-394

Our results are also fully consistent with previous measurements done on the system. In D’Angelo et al., it was shown that apparent tissue elasticity increases with time when cells are probed using small beads located at the apical surface. We show that this apparent drop in elasticity coincides with the time when cells grow sufficiently tall for the bead to no longer contact the basal side. In this way, our interpretation is in complete agreement with previously reported data (although not with the previously proposed interpretation of that data). This is extensively discussed in the Discussion section (lines 474-496).

Additionally, our conclusion is supported by the well-established distribution of F-actin. Our previous data (Doubrovinski et al. PNAS 2017) showed that tissue elasticity is overwhelmingly dominated by cortical F-actin. It is well established (e.g. Sokac and Wieschaus JCS 2008) that basal side exhibits much stronger F-actin signal as seen from phalloidin staining than either the apical or the lateral side. These data together strongly suggest that basal domains are much stiffer than both the lateral and the apical domains in complete accordance with our key conclusion. We have now added a discussion of this point (lines 497-502).

To address the point about the robustness of model predictions, we have done additional simulations where cells were sealed on their basal side to see if this modified geometry could explain observations without assuming that different cellular domains have different stiffness. We found that the value of spring stiffness must be chosen between 0.001 and 0.01 to match the measured rate of tissue recoil (corresponding to tau=120 s and tau=40 s respectively, compared to the experimental value of 60 s). In these cases, the predicted maximal force during loading would be between 0.26 nN and 1.3 nN, which is about an order of magnitude too small compared with the experimentally measured value. This is the same discrepancy that we saw with our initial model where basal sides were open, indicating that this result is not due to the specific assumption of open basal sides. These data further supports that our model is robust with respect to varying parameters and some aspects of the assumed tissue geometry.

8) The authors reported measurements without error bars or units in several places throughout the manuscript. For example, on page 7, there should be error bars for the cantilever Young modulus measurements or E on top of page 12 or k in line 350. The authors should go through these issues and report appropriate numbers.Similarly, the name *Drosophila* should be in italics throughout the text (it is mixed now).

We apologize for this oversight. The corresponding changes have now been made. However, we were not able to produce error bars for the reported values of the elastic constants estimated from simulations: as was mentioned in the text, our numerical simulations are extremely laborious, requiring approximately 3 weeks to run. This makes sensitivity analysis prohibitively time-expensive, so we report the estimated constants without giving the error bars. Also, we emphasize and stress that these constants must be considered order of magnitude estimates, and not precisely determined values. We included a note on this (lines 392-394). However, to the best of our knowledge no (semi)-quantitative model of the same predictive power describing *in-vivo* tissue rheology has been reported in the past.